# A measurement system for $CO_2$ and $CH_4$ emissions quantification of industrial sites using a new in situ concentration sensor operated on-board Unmmanned Aircraft Vehicles

Jean-Louis Bonne[1], Ludovic Donnat[2], Grégory Albora[1], Jérémie Burgalat[1], Nicolas Chauvin[1], Delphine Combaz[1], Julien Cousin[1], Thomas Decarpenterie[1], Olivier Duclaux[2], Nicolas Dumelié[1], Nicolas Galas[2], Catherine Juery[2], Florian Parent[1], Florent Pineau[2], Abel Maunoury[2], Olivier Ventre[2], Marie-France Bénassy[2], Lilian Joly[1]

[1]GSMA UMR 7331, Univerisité de Reims Champagne-Ardenne, CNRS, Reims, 51100, France
[2]Air Quality Laboratory, TotalEnergies R&D, Solaize, 69360, France

*Correspondence to*: Lilian Joly (lilian.joly@univ-reims.fr), Jean-Louis Bonne (jean-louis.bonne@univ-reims.fr)

**Abstract**

We developed and tested a complete measurement system to quantify $CO_2$ and $CH_4$ emissions at the scale of an industrial site based on the innovative sensor Airborne Ultra-light Spectrometer for Environmental Application (AUSEA), operated on-board Unmanned Aircraft Vehicles (UAVs). The AUSEA sensor is a new light-weight (1.4 kg) open path laser absorption spectrometer simultaneously recording in situ $CO_2$ and $CH_4$ concentrations at high frequency (24 Hz in this study) with precisions of 10 ppb for $CH_4$ and 1 ppm for $CO_2$ (when averaged at 1 Hz). It is , suitable for industrial operation at a short distance from the sources (sensitivity up to 1000 ppm for $CO_2$ and 200 ppm for $CH_4$). Greenhouse gases concentrations monitored by this sensor throughout a plume cross section downwind of a source drive a simple mass balance model to quantify emissions from this source.

This study presents applications of this method to different pragmatic cases representative of real-world conditions for oil and gas facilities. Two offshore oil and gas platforms were monitored for which our emissions estimates were coherent with mass balance and combustion calculations from the platforms. Our method has also been compared to various measurement systems (gas LiDAR, multispectral camera, infrared camera including concentrations and emissions quantification system, acoustic sensors, ground mobile and fixed Cavity RingDown Spectrometers) during controlled release experiments conducted on the TotalEnergies Anomaly Detection Initiatives (TADI) test platform at Lacq, France.It proved suitable to detect leaks with emission fluxes down to 0.01 g s$^{-1}$, with 24 % of estimated $CH_4$ fluxes within the -20 % to +20 % error range, 80 % of quantifications within the -50 % to +100 % error range and all of our results within the -69 % to +150 % error range. Such precision levels are better ranked than current top-down alternative techniques to quantify $CH_4$ at comparable spatial scales.

This method has the potential to be operationally deployed on numerous sites and on a regular basis to evaluated the space and time dependent greenhouse gases emissions of oil and gas facilities.

## 1 Introduction

After $CO_2$, methane is currently the second most important anthropogenic greenhouse gas in terms of climate forcing (Etminan et al., 2016), with effective radiative effects between 1750 and 2019 of $0.54\pm0.11$ W m$^{-2}$ for CH$_4$ compared to $2.1\pm0.26$ W m$^{-2}$ for CO$_2$ (Forster et al., 2021). Methane was brought to the centre of the political debate, with new pledges of parties to consider further actions to reduce non-carbon dioxide greenhouse gas emissions by 2030 (Glasgow Climate Pact | UNFCCC, 2021). Due to its short lifetime of $11.8\pm1.8$ years in the atmosphere (Forster et al., 2021), reducing CH$_4$ emissions would be effective in terms of climate mitigation on short timescales (Shindell et al., 2012): fossil CH$_4$ emissions have a global warming potential of $82.5\pm25.8$ over 20 years, but of $29.8\pm11$ over 100 years, in comparison with CO$_2$ with reference global warming potential of 1.0 (Forster et al., 2021). Climate mitigation actions including fast and deep methane emissions reduction would limit climate overshoot linked with concomitant decrease of climate cooling aerosols emissions (Masson-Delmotte et al., 2018). Large uncertainties exist in the variations of many methane anthropogenic and natural sources and sinks (Saunois et al., 2020). A recent study indicates that anthropogenic fossil CH$_4$ emissions may have been underestimated by about 25 to 40 %, representing about 38 to 58 Tg CH$_4$ per year (Hmiel et al., 2020).

According to inventories, Oil and gas (O&G) sector would be responsible for 22 % of the global anthropogenic methane emissions (Saunois et al., 2020). O&G facilities can emit methane from multiple sources (high elevation stacks and flares; common or local vents; fugitive sources) of different nature (process venting; incomplete combustion during flaring, power generation, heating, etc; unintentional leaks) (Oil and Gas Methane Partnership (OGMP) 2.0 Framework, 2022). O&G operators currently report their methane emissions to their stakeholders, based on calculations using bottom-up approaches (Ng et al., 2017), including flow meters inside the plant, emission factors, modelling and Leak Detection And Repair (LDAR) campaigns. Such methods hardly capture temporal variations of emissions, unexpected operations and are furthermore poorly adapted to fugitive or diffuse emissions. This is an important issue as recent estimates suggested that fugitive emissions represent a significant part of emissions from O&G activities and could be strongly underestimated (Alvarez et al., 2018). Fugitive emissions might have been increasing in recent year, which would partly explain the global methane atmospheric concentrations increase observed since the mid-2000s (Worden et al., 2017).

Top-down approaches, based on atmospheric measurements, can complement and validate bottom-up flux estimates. Developing technics able to be implemented on industrial facilities are necessary, either for fast leak detection or for quantification of long-term greenhouse gases emissions. They should be validated via controlled release experiments, which can be organized within intercomparison campaigns (Ravikumar et al., 2019; Feitz et al., 2018). Such controlled release campaigns are for example organized yearly on the TotalEnergies Anomaly Detection Initiatives (TADI) infrastructure in Lacq, southwestern France (43.41°N, -0.64°W), an industrial area dedicated to the simulation of a real-size oil and gas facility, used by international groups to validate their emission detection or quantification techniques (Kumar et al., 2021; Druart et al., 2021).

At the facility scale, different top-down emissions quantification approaches already exist, relying on both in situ and remote sensing measurements. Some methods, well adapted to emissions quantification on flat terrains such as landfills, like eddy covariance, stationary mass balance methods, radial plume mapping (Mønster et al., 2019), cannot be adapted to all industrial contexts with complex topography and high elevation sources. In situ atmospheric concentration measurements at the surface can be obtained from analysers at a fix position or on

mobile platforms such as in cars for onshore facilities (Brantley et al., 2014; Ars et al., 2017; Feitz et al., 2018; Yacovitch et al., 2020; Kumar et al., 2021) or on-board ships for offshore facilities (Nara et al., 2014; Riddick et al., 2019; Yacovitch et al., 2020). Other methods based on airborne observations have the advantage of measuring concentrations directly inside the plume. Observations can be performed from aircraft for onshore (Terry et al., 2017; Hirst et al., 2013; Lee et al., 2018; Conley et al., 2016, 2017; Gorchov Negron et al., 2020) or offshore facilities at the scale of an individual platform or of a whole basin (Gorchov Negron et al., 2020; France et al., 2021; Fiehn et al., 2020), but with a high logistical and financial cost, and at a long distance from sources. The choice of the type of mobile platform is a key parameter, as it will determine the speed at which measurements are performed and difficulties may appear with low speed platforms, for example when the plume is changing direction over the monitoring period, or if areas cannot be accessed (limitations by the road infrastructure for cars, minimum distance to the facilities of minimum flight elevations for aircraft measurements, observations restricted to low elevations for cars or ships). UAV-based observations are adapted to the scale of industrial facilities, including offshore, and might answer these challenges. UAVs may indeed operate at lower costs than aircraft. They provide high speed and reactivity, allowing to fly at shorter distances from the sourcescompared to aircraft or boats. This facilitates validation by controlled release experiments and induces a gain in sensitivity as dilution of effluents will be lower at short distance from the source. The possibility to fly inside industrial sites also permits to better localize the emission sources.

For quantifying emission fluxes based on airborne concentration measurements, two main approaches are generally adopted. The first approach is based on the inversion of modelled Gaussian plumes (Hirst et al., 2013; Lee et al., 2018; Shah et al., 2020). The Gaussian-based inversion methods are commonly applied to ground mobile observations (Brantley et al., 2014; Kumar et al., 2021) or to localize multiple unknown sources (Hirst et al., 2013; Huang et al., 2015; Brereton et al., 2018). Recent UAV-based experiments relied on a near-Gaussian inversion approach but so far suffer from important uncertainties (Shah et al., 2020), which might be improved in future (adapted measurement protocol or quantification model). The second approach is a mass balance method consisting in comparing the fluxes of gas entering and exiting a box around a source. It does not rely on any atmospheric model but is a direct quantification of the flux based on its integration through a surface. The main difficulties associated with this method are of being able to measure the concentrations throughout the whole plume and of having a precise knowledge of the wind conditions. This type of approach was originally employed for DIAL (DIfferential Absorption LiDAR) quantifications, providing state-of-the-art Volatile Organic Compounds (VOC) quantification in complex industrial plant (NF EN 17628, 2022), and was already applied to greenhouse gases emissions quantification at various scales from industrial sites to large cities based on UAV or aircraft observations (Mays et al., 2009; Karion et al., 2015; Nathan et al., 2015; Allen et al., 2019; Fiehn et al., 2020; Morales et al., 2022). Contrary to Gaussian-based inversion models, mass balance does not require the assumptions of constant and continuous emissions creating a steady-state system with normally distributed pollutant concentrations over a flat and uniform terrain, which is often none applicable to onshore or offshore fields.

Identification and quantification of $CO_2$ and/or $CH_4$ sources via top-down UAV-based approaches atmospheric concentration measurements of these species with a very low response time and high frequencies are preferable to be able to detect small plumes with UAVs flying at elevated speeds. Required precisions and sensitivity ranges

depend on the expected amplitude of the signal. For applications to oil and gas industries, a large range of measurable concentrations is required, as high concentrations above the background level are expected, therefore the precision might be low as long as there is a good signal to noise ratio. Accurate analysers are not required but their linearity is important since relative concentrations compared to the background levels are used. Different types of methane sensors suitable for UAV-sampling already exist. Metal oxide gas sensors (Neumann et al., 2013; Malaver et al., 2015; Liu et al., 2020; Rivera Martinez et al., 2021) or cryptophane-A cladded Mach-Zehnder interferometers (Dullo et al., 2015) are compact and competitive in price but with a relatively high detection limit and low response time (17 ppm of $CH_4$, 10 s response time) (Dullo et al., 2015). Miniaturized laser-based sensors also emerged in the last years (Berman et al., 2012; Khan et al., 2012; Golston et al., 2017, 2018; Nathan et al., 2015; Shah et al., 2020; Tuzson et al., 2020), but do not necessarily have a large sensitivity range, a low response time and a light weight below 2 kg and generally measure only one species. Tunable Diode Laser Absorption Spectroscopy (TDLAS) allows a high selectivity and sensitivity in the gases detection and is considered as the most advantageous technique for measuring atmospheric gas concentrations (Durry and Megie, 1999). Many applications are already based on this technique, not only UAV applications, among which Cavity Ring-Down Spectroscopy (CRDS) (Crosson, 2008; Chen et al., 2010; Rella, 2010), Cavity Enhanced Absorption Spectroscopy (CEAS) (Romanini et al., 2006), Integrated Cavity Output Spectroscopy (ICOS) (O'Keefe, 1998; O'Keefe et al., 1999) or the most straightforward Direct Absorption Spectroscopy (DAS) (Xia et al., 2017). Compared to close path DAS, open cavity instruments are more sensitive to environmental perturbations, such as temperature and pressure variations or perturbations by solar radiations, but they have the advantage of a substantially enhanced response time to concentrations changes. They also do not require pumps or cell temperature and pressure regulation systems, sparing substantial weight and energy. DAS is well adapted to in situ measurements and can be applied to sensors light enough to be embarked on UAVs, which led to the choice of technology adopted for the development of the sensor presented in this study.

In this study, we present a newly developed UAV-embarked $CO_2$ and $CH_4$ in situ analyser and a methodology of emissions quantification adapted to the monitoring of O&G facilities. We present the characterization of this analyser for the environmental conditions of its field applications. Our emissions quantification method has been validated against $CH_4$ controlled releases in an intercomparison effort during the TADI campaigns of 2019 and 2021, together with other quantification methods using varied technologies: multispectral camera, ground based CRDS (fix stations or mobile measurement in a car), wind and gas LiDAR, infrared camera including concentrations and emissions quantification system, or Tunable Diode LiDAR. As a large part of TotalEnergies production activities are offshore-based, we present an application of our method to the quantification of emissions of two offshore gas production platforms in the North Sea.

## 2   $CO_2$, $CH_4$ and $H_2O$ analysers for UAV in situ observations

### 2.1   Technical description

A new sensor has been developed for in situ $CO_2$ and $CH_4$ observations able to operate on-board UAVs (see Figure 1): the Airborne Ultra-light Spectrometer for Environmental Application (AUSEA). It is based on the technical concept of the AMULSE instrument (Joly et al., 2016, 2020). As for the AMULSE instrument, the AUSEA instrument includes an open-path infrared Laser absorption spectrometer using two DFB interband cascade laser

diodes in the mid-infrared spectral region (NIR): near 4 µm with a direct path of 11 cm to measure $CO_2$
concentrations and near 3 µm in a home-made Herriott multipass cell of 3.5 m path length to measure $CH_4$
concentrations. The measurement frequency is of 24 Hz.

Compared to the AMULSE instrument, the AUSEA instrument has been adapted to reduce its weight, to adapt its
sensitivity range to industrial applications (up to 1000 ppm in $CO_2$ and up to 200 ppm in $CH_4$), to limit the effect
of vibrations, air turbulences, magnetic perturbations and to implement air-ground communication for a real time
visualisation of the concentrations by the operators. It has a power consumption of 8 to 15 W in most usual cases,
depending on the external temperature (with maximal power consumption of 30 W during less than 1 s at start-
up). It can be powered either with dedicated batteries for an average lifetime of 1.5 hours or directly by the UAV.
The instrument is also equipped with an IMET 4 from InterMet Systems (modified to fit in the instrument) to
record air temperature (repeatability of 0.2°C, response time at 2s for still air and <1 s at 5 m.s-1 and 1000hPa),
pressure (response time of 0.5 ms, 1.0 hPa uncertainty, 0.75 hPa reproducibility) and relative humidity (response
time of 0.6s at 25°C or 5.2s at 5°C and repeatability of 5 %) at 1Hz frequency. Temperature, pressure and humidity
values measured by the iMet-4 are used by the inversion process to account for their spectroscopic effects on the
$CO_2$ and $CH_4$ absorption lines. The humidity values measurement by the iMet-4 can be employed to calculate the
concentration in dry air, considering the dilution effect of the water vapour.

To monitor the position of the instrument, a LiDAR Lightware LW20/C measures the distance to the ground and
an Adafruit GPS for records position and time. Position data obtained from the UAVs themselves have also been
used for post processing as they have a better precision than the integrated GPS sensor. Altogether, the weight of
the AUSEA sensor has been optimized down to 1.4 kg, including all previously listed hardware. The results
presented in this study are based on experiments performed with two AUSEA instruments (hereafter named
AUSEA111 and AUSEA112), in order to verify the reproducibility of performances between several analysers.
Laboratory tests were performed to evaluate the precision and stability of the instruments in controlled
environments. Field applications are also presented, using these instruments and will also be analysed in terms of
instrumental performances (analysing in-flight precision).

### 2.2    In-lab $CO_2$ and $CH_4$ analysers characterisation

In-lab characterisation of the stability and linearity was performed independently on AUSEA111 and AUSEA112
instruments, and repeated at different periods in 2021 and 2022. For these experiments, each AUSEA instrument
was placed in a custom-made atmospheric chamber in which air is continuously mixed and homogenised (using
fans) and temperature is regulated (at laboratory temperature).

#### 2.2.1    Stability

The stability experiments consisted in measuring the same air sample within the closed atmospheric chamber by
the AUSEA instrument over several hours. For AUSEA112, two experiments are analysed (conducted on 2022-
04-19 for a duration of 3 hours and 2 minutes and of 1 hour and 13 minutes); while four experiments are analysed
for AUSEA111 (two were conducted on 2022-06-08 for respective durations of 1 hour and 35 minutes and of 15
hours and 12 minutes and two were conducted on 2022-03-23 for respective durations of 50 minutes and 1 hour
and 50 minutes). Allan deviation, calculated from those experiments for both analysers, are presented in Figure 2
and Table 1. The precision of our measurements can be derived from these experiments: for $CH_4$, precisions are

below 20 ppb at 2 Hz, below 10 ppb at 1 Hz, below 1 ppb at 10 s and below 0.2 ppb at 1 minute; for $CO_2$, precisions are below 2 ppm at 2 Hz, below 1 ppm at 1 Hz, below 0.1 ppm at 10 s and of 0.01 ppm at 1 minute. We note a minimum of precision for the instrument AUSEA112 at 60 seconds with a stagnation of performances for longer averaging periods, contrary to the instrument AUSEA111 which has a better longer-term stability. For comparison, precisions (1σ) of a commercial Picarro Inc. model G2401 analyser, are of 0.05 ppm for $CO_2$ and 0.5 ppb for $CH_4$ at 5 seconds, but this type of analyser has very different applications from our sensor (weight of 26.9 kg). Other laser-based UAV-embarked technologies have reached precisions at 1 Hz for $CO_2$ of 0.6 ppm (Berman et al., 2012) and for $CH_4$ of 1 ppb (Tuzson et al., 2020), 2 ppb (Berman et al., 2012; Shah et al., 2020), 5 ppb (Golston et al., 2017) or 100 ppb for $CH_4$ (Nathan et al., 2015), but with various weights, power consumption or response times.

### 2.2.2    Linearity

To evaluate linearity, air samples of varying concentrations were simultaneously measured by the AUSEA analyser placed in the atmospheric chamber and by a reference instrument pumping air from the atmospheric chamber. An air with high $CH_4$ concentration was initially injected in the atmospheric chamber and progressively mixed with room air, thus spanning a continuous range of concentrations from the initial sample up to ambient air levels. Variations of $CO_2$ concentrations were simply generated by natural variations of the $CO_2$ values in the laboratory air. The reference instrument used was a Cavity Ring-Down Spectrometer (Picarro Inc. model G2401), hereafter referred as Picarro, with an operating range certified by the manufacturer from 0 to 1000 ppm for $CO_2$ and from 0 to 20 ppm for $CH_4$. The Picarro has been validated through the ICOS Atmospheric Thematic Center protocol (Yver Kwok et al., 2015) and was calibrated using the standard procedure for ICOS atmospheric monitoring stations with 4 calibration standards of known $CO_2$ and $CH_4$ concentration ranging from 396.05 to 504.16 ppm for $CO_2$ and 1807.7 to 2346.5 ppb for $CH_4$ (ICOS RI, 2020). AUSEA and Picarro analysers data were compared at the Picarro temporal resolution of 5 seconds. Linearity experiments were conducted on 2022-03-23 and 2022-06-08 for AUSEA111 and on 2021-04-15 and 2022-04-19 for AUSEA 112. Linearity experiment covered $CO_2$ and $CH_4$ concentrations ranging from 429.0 to 861.4 ppm of $CO_2$ and from 2.1 to 20.00 ppm of $CH_4$ (within the reference instrument certified linearity domain).

The results of the linearity experiments are presented on Figure 3 and Table 2. An excellent linearity was observed for both species for each experiment: linear regressions provide excellent coefficients of determination $R^2$ of 1.0 for $CH_4$ and $CO_2$, with p-values (probability of obtaining tests results at least as extreme as the results actually observed) well below $10^{-5}$, so with high statistical validity. Low residuals are observed for each linear regression (difference between measured values and linear regressions): within 0.02 ppm of $CH_4$ and 1.5 ppm of $CO_2$ (Figure 3), which corresponds to the precisions of the instruments and do not reveal deviations from a linear distribution. We observed relatively low variations of the slopes and intercepts of the linear regressions between repeated experiments over the course of several months (Table 2), therefore of the instrument response (slopes and intercepts variations respectively below 2.3 % and 0.16 ppm for $CH_4$ and 1.6 % and 7 ppm for $CO_2$).

The linearity of our AUSEA sensor was experimentally validated for $CO_2$ concentrations between 429.0 and 861.4 ppm and $CH_4$ concentrations between 2.1 and 20 ppm, against the guaranteed linearity domain of a reference instrument validated top-of-the-art metrology standards (Yver Kwok et al., 2015) . However, the sensitivity domain of our AUSEA sensor exceeds these limits: the chosen pathlength for the $CH_4$ measurements, has been

determined to reach saturation around 200 ppm. Given the saturation of the $CO_2$ absorption spectrum, the maximum of measurable concentration is limited to 1000 ppm (but this limit can be easily adapted by modifying the $CO_2$ laser-to-detector pathlength). Therefore, we believe the linearity domains also exceed the range of concentrations tested in the laboratory, up to 1000 ppm for $CO_2$ and above 100 ppm for $CH_4$. The lack of a reference instrument with a comparable certified linearity domain in our laboratory did not allow us to validate this limit so far. However, additional linearity experiments conducted with the same reference CRDS instrument, not presented here, for $CH_4$ concentrations up to 100 ppm also depicted an excellent linearity (also with $R^2$ of 1.0 and $p < 10^{-5}$ for 24975 data points), therefore giving confidence in the linearity of our AUSEA sensors, even for concentrations out of the CRDS instrument manufacturer's certified linearity domain. This confidence is also motivated by the fact that the same type of CRDS analysers were also employed for the quantification of industrial emissions of $CH_4$ with peaks up to approximately 90 ppm (Kumar et al., 2021; Jackson et al., 2014).

## 3    Source emissions quantification

A mass balance method has been developed to quantify source emissions from atmospheric concentration measurements. It relies on the airborne monitoring of atmospheric concentrations of the species of interest from UAV and of the wind speed and direction at the elevations of the UAV.

### 3.1    Monitoring method

#### 3.1.1    Measurements on-board Unmanned Aircraft Vehicles

The AUSEA instrument is embarked on a low-weight (below 8 kg payload) commercial multicopter. Several models of UAVs have been employed (DJI M200, DJI M210, DJI M300, and a non-commercial drone), able to flight under wind speeds up to 12 m s$^{-1}$, with autonomies of 20 to 45 minutes. The instrument was always integrated between both UAV landing gears, below the propellers level (see Figure 1). Concentration measurements are remotely monitored in real-time by the operators on the ground (usually a pilot and a co-pilot), allowing to locate the plume and optimize the trajectory of the UAV to fit to the flight plan requirements of the emissions quantification method.

#### 3.1.2    Wind profiles meteorological parameters measurements

Wind speed and direction profiles are recorded by a commercial ZX300 Doppler wind LiDAR (from ZX LiDARs Inc.), equipped with an AIRMAR weather station at 2.5 m above ground level (or m.a.g.l.). The LiDAR records wind speed and direction at 10 elevations between 11 and 300 m.a.g.l., completed by wind measurements at the AIRMAR station, thus covering the range of altitudes of the UAV tracks. The AIRMAR station also records temperature, relative humidity and air pressure. Wind measurements have an approximate 15 to 20 s time resolution (all levels are monitored successively within about 1 to 2 s), with precisions of 0.1 m s$^{-1}$ for the wind speed (WS) and of 0.5° for the wind direction (WD). The wind speed and direction are interpolated at the elevations of the UAV. For elevations below the first height of LiDAR measurements, a logarithmic interpolation with

assumption of null wind speed at the ground level is used, following the shape of a neutral wind profile. For levels above the first LiDAR measurement height, the interpolation is linear.

### 3.1.3 UAV flight protocol

Our protocol for UAV-based atmospheric concentrations monitoring was designed for our quantification model. The UAV flight plan should meet the conditions described hereafter (see Figure 4). Concentration measurements are performed downwind of the sources, within a vertical plane crossing the plume, later referred as the observational plane. The observational plane must be as close as possible to a plume cross-section, therefore orthogonal to the prevailing wind direction. Several horizontal transects covering the entire plume and part of the surrounding background are recorded within this plane, with elevations distributed from below (or closest to the ground possible) to above the plumes. A precise wind speed and direction monitoring covering the range of altitudes of the UAV must be conducted simultaneously.

### 3.2 Emissions quantification model

An emission quantification approach has been tested to take advantage of the UAV observations. It is based on a mass balance approach, as also applied in the literature for similar UAV-based flight scenarios (Yang et al., 2018; Andersen et al., 2021; Morales et al., 2022): the emission rate $Q$ (in g s$^{-1}$) is estimated from a flux through the observational plane crossing the plume of emissions. It assumes constant emissions during the monitoring period and no degradation of effluents through chemical reactions over the monitoring period, which is reasonable for $CO_2$ and $CH_4$. The referential $x, y, z$ is defined by the observational plane (see Figure 4), with $x$ in the horizontal direction orthogonal to the plane, $y$ in the horizontal direction along the plane and $z$ in the vertical direction. $Q$ is equal to the integral across the plane, of the wind speed component along $x$, $u_x(y,z)$ (in m s$^{-1}$) multiplied by the differential of dry basis volume concentrations (in g.m$^{-3}$, calculated from wet basis concentrations and the humidity measured by the iMet-4 sensor) between the plume $c_p(y,z)$ and the background $c_{bg}(y,z)$:

$$Q = \iint_{y,z} u_x(y,z) \cdot [c_p(y,z) - c_{bg}(y,z)] \, dydz \,, \tag{1}$$

Background concentrations are assumed spatially uniform, $c_{bg}(x,y,z) = c_{bg}$ and estimated from the concentrations measured outside the plume. Wind speed is assumed horizontally uniform: $u_x(y,z) = u_x(z)$. As the wind direction might fluctuate over the complete monitoring period, we consider the average wind over the duration of each transect. Noting $\alpha(z)$ the angle, often non-neglectable in practice, between the wind direction and the orthogonal to the transect, the component $u_x(z)$ of the wind speed can be expressed as a function of the total wind speed $U(z)$, as follows: $u_x(z) = \cos(\alpha(z)) \cdot U(z)$. Altogether, Eq. (1) becomes:

$$Q = \int_z U(z) \cdot \cos(\alpha(z)) \cdot \left[ \int_y (c_p(y,z) - c_{bg}) \, dy \right] dz, \tag{2}$$

For the computation based on observations data, we first calculate the values of $q(z) = U(z) \cdot \cos(\alpha(z)) \cdot \left[ \int_y (c_p(y,z) - c_{bg}) \, dy \right]$, the horizontal flux component (in g.s$^{-1}$.m$^{-1}$) at each transect level $z$. The high horizontal resolution of the measurements allows using a simple linear integration to compute the horizontal integration. The integral of all $q(z)$ values along $z$ is calculated from interpolated values of $q(z)$ profile, assuming neglectable vertical variations of the plume compared to the vertical gap between successive transects. Linear vertical interpolations are used between the values of $q(z)$ at each horizontal transect elevation $z$. If values of $q(z)$ do not

reach zero at the lowest or highest horizontal transect elevations, an extrapolation is performed using a logarithmic function, assuming that $q(z)$ must be equal to zero at the ground. This method can be applied in a wide range of meteorological conditions (limited by UAV maximum wind speeds limits), but is poorly adapted to low wind speeds and unstable wind directions, where measurement uncertainty can strongly rise (Yang et al., 2018; Morales et al., 2022).

### 3.3 Validation of emissions quantification method

### 3.3.1 Field validation protocol

Two validation campaigns were conducted from 1 to 10 October 2019 and from 07 to 10 September 2021 on the TotalEnergies Anomaly Detection Initiatives (TADI) platform in Lacq, in southwestern France (43.41°N, -0.64°W). The TADI platform, already described in the literature (Kumar et al., 2021, 2022), is an approximately 2000 m² almost flat rectangular area (Figure 5), surrounded by agricultural land and rural settlements, and important chemical and industrial plants on the east of the platform. Multiple obstacles for dispersion are created by tents where other instruments are located, decommissioned oil and gas equipment and other small infrastructures. A road surrounding the north and east borders of the site cannot be flown over, limiting the area of UAV operations.

Several emission sources were spread over the platform, within a 40x60 m rectangular area classified as "ATEX zone" (Figure 5), out of reach for all participants due to security reasons. Sources were elevated between 0.1 and 6.5 m.a.g.l, originating from a variety of equipment (valve, connector, flange, drilled plug, tank, manhole, corrosion, flare pipe - no combustion, etc.). Either $CO_2$ or $CH_4$ or a combination of both species were emitted, but also a mixture including a proportion of $C_2H_6$ or $C_3H_8$, to test if the method is able to differentiate these species from $CH_4$. Only $CH_4$ emissions quantifications results are presented in this study (the low number of $CO_2$ releases does not allow statistical analysis of our $CO_2$ quantifications). Release scenarios had durations from 10 to 73 minutes (with two short-lasting leaks of a 15 seconds and 2.5 minutes which were not be monitored with our method), with pauses of approximately 5 minutes between two releases. Mass flow controllers were used to regulate and monitor the controlled $CH_4$ flow rates, with a large range of values from 0.01 to 150 g s$^{-1}$. This variety of emission sources, duration and amplitude is representative of the diversity of emission scenarios expected on industrial facilities. Information about the leaks (locations, species and fluxes) of each experiment can either be communicated (open trials) or withheld (blind tests) from the measurement teams. Results from both open trials and blind tests are presented.

For our UAV-based emission quantification method, one team was operating a DJI M200 in 2019, while two teams were operating either a DJI M300 or a DJI M210 and a non-commercial UAV in 2021. $CO_2$ and $CH_4$ concentrations were measured on-board these UAV with an AUSEA analyser. (either AUSEA 111 or AUSEA112).

In 2021, all drones were equipped with RTK GPS positioning systems, which was not the case in 2019. Flight durations have been from 10 to 20 minutes. Concentration measurements were performed within a vertical plane distant from the sources from approximately 20 to 80 m. As the sources were at low elevation (below 6.5 m), the plumes were monitored with a varying number of 5 to 15 low elevation horizontal transects distributed between 1 and 12 m.a.g.l. in 2019 and up to 35 m.a.g.l in 2021. Wind speeds and directions were measured at 10 elevations between 11 and 300 m.a.g.l. with the ZX300 wind LIDAR (equipped with the AIRMAR station at 2.5 m.a.g.l.).

### 3.3.2 Results of validation experiments

An example of concentrations measurements obtained during the TADI 2021 is presented in Figure 6. Spikes of $CH_4$ concentrations up to 15 ppm linked to the emission plume can be observed well above the background level (around 2.1 ppm for this flight). The vertical distribution of the concentration measurement shows the highest spikes along a transect around 7 m.a.g.l.

Figure 7 presents the horizontal components of the flux $q(z)$ at each horizontal transect elevation and the vertical interpolation of these values used for the computation of the vertical integration. The vertical profile of $q(z)$ shows a peak around 7 m elevation. Null values of q(z) have been correctly measured around 18 m elevation, describing the top of the plume. Low values of q(z) are obtained at the lowest horizontal transect but the bottom value does not reach zero (the lowest horizontal level is determined by safety reasons of UAV operation). Therefore, a logarithmic interpolation has been performed to interpolate the values of q(z) below the lowest transect, considering zero flux at the ground level. $CH_4$ emissions quantifications of the two TADI campaigns are analysed hereafter and compared to the reference real fluxes derived from mass flow meters at the source. The emissions averaged quantifications for each controlled release experiment are presented on Figure 8 and the values are given in the Supplementary Materials on Tables S1 and S2 (with additional details for each flight). Statistical analyses of the results are presented in ,Table 3 and Table 4.

During the two TADI campaigns, UAV measurements were conducted during 34 out of 41 controlled releases (among which 15 were blind tests) in 2019 and during 20 out of 24 controlled releases (all blind tests) in 2021. Emission quantifications could be successfully calculated with our method for respectively 26 and 18 controlled release experiments in 2019 and 2021. Some release experiments could not be quantified due to unavailability of the instruments, UAVs or pilots and some of the quantification flights were discarded as the flight paths did not match our standards (e.g. did not cover the complete horizontal or vertical plume section or technical issues were noticed with some of the sensors). Some of the controlled releases could be monitored by several independent flights (3 by 4 flights, 4 by 3 flights, 19 by 2 flights) and the rest (19 releases) could only be monitored once.

Figure 8 and Table S1 and S2 present the averaged quantifications of all controlled release experiments compared to the real fluxes. The relative errors of the average of quantifications are also given in Table S1 and S2. The relative errors of our quantification compared to the true values show that out of 45 quantified controlled releases, 24 % relative errors between -20 and +20 % compared to the true values (11 out of 45 controlled releases, cf. Figure 8), and 80 % of our quantifications had relative errors between -50 % and +100 % (36 out 45 controlled releases, cf. Figure 8). Among all 45 quantifications of the TADI campaigns, the relative errors of all quantifications span between -69 % and +149 % (Table 3), which is representative of the global precision of the quantification method. In terms of total emitted mass during the whole TADI campaigns, the relative error of our quantifications is of -41 %, taking into account all the quantified releases, but this total is strongly biased by the highest release 2019-W41-SAT at 150 g.s$^{-1}$ during 01 hour 13 minutes. By discarding values associated to this release, the relative error of our quantifications is of -12 % only.

### 3.3.3 Sources of uncertainties of our emissions quantification approach

This mass balance approach could integrate a direct computation of uncertainties based on the propagation of measurements uncertainties from the wind and from the concentrations, but these calculations where not integrated in the currently developed algorithm. This will be the subject of further developments. However, the uncertainties of the mass balance approach are not limited to the instrumental uncertainties. Other sources of uncertainties are associated (i) with the vertical interpolation between the successive horizontal transects where concentration measurement are performed, as information is missing between these levels, (ii) and with the turbulent nature of atmospheric transport and the fact that the measurements do not represent either an instantaneous picture nor an average situation of the plume (the monitoring of the concentrations with the UAV is performed along a trajectory within several minutes during which the plume changes, and the wind profile is measured at a low-frequency and at a non-neglectable distance from the UAV flight plan). These sources of uncertainties are more difficult to estimate than the propagation of instrumental uncertainties. The wind variability can be estimated based on measurements, but the effect of the vertical interpolation is more difficult to estimate. The orders of magnitudes of uncertainties associated with the concentration and with the wind are analyzed hereafter.

In-flight performances of the AUSEA instruments may be different from the performances achieved during the laboratory experiments under controlled temperature and pressure conditions without any mechanical perturbations due to the vibrations of the UAV. The in-flight accuracies of our measurements cannot be estimated from these experiments as there was no reference instrument. We evaluate the precision of the concentration measurements for each flight as the median value of the 1 second rolling standard deviation (median value is being used to avoid the influence of the real measured spikes within the plumes). These values are given in Table S1 and S2 for $CH_4$ and $CO_2$. The precision for all flights on the $CH_4$ concentrations measurements is always below $\pm 0.3$ ppm. The median value is of the in-flight $CH_4$ precision at 1 second was of 118 ppb in 2019 and of 30 ppb in 2021. This important difference on the precision of the $CH_4$ measurements depict strong improvements of the analyzers 395 between the 2019 and the 2021 campaigns. For $CO_2$, values of each flight are not shown in Tables S1 and S2, but the median value of the in-flight $CO_2$ precision at 1 second reaches 0.3 ppm in 2019 and 0.2 ppm in 2021. Uncertainties linked with the concentration may result both from the instrumental precision and from the amplitude of the signal (signal to noise ratio). We calculated the signal to noise ratio considering the signal as the amplitude of the measured concentrations and the noise as the median value of the 1 second rolling standard deviation. Compared to the signal level on the order of magnitude of 10 to $10^2$ ppm for $CH_4$, the measurement uncertainties on the order of magnitude of $10^{-1}$ to $10^{-2}$ ppm, the signal to noise ratios of each flight, presented on Tables S1 and S2, are always good, ranging from 38 to 8017, with a median value of 618. As presented on Figure S1, our result show that the envelop of the distribution of quantification errors follows a decreasing trend with increasing signal to noise ratios. The highest quantification errors of 182 % correspond to a signal to noise ratio of 132, while the flight with the highest signal to noise value of 8017 reaches quantification error of 25 %. This would indicate that the signal to noise ratio is an indicator of the quality of the quantification.

One could expect the signal to noise ratio to be determined by the flow of emitted $CH_4$ and thus having a link between the performances of the quantification and the emitted flow: Table 4 presents a classification of the quantifications in terms of performance classes, for different ranges of real $CH_4$ emission fluxes. However, the

performances of our quantifications do not show any trend depending on the emissions (Table 4). In fact, the signal to noise ratio is not related to the emitted flow, as the distance of the flight plan and the source can be different for each flight.

A comparable study using mass balance emissions quantification with UAV-based measurements (Andersen et al., 2021) noticed that the wind was the dominant source of uncertainty and that the final uncertainty of the quantification could be assimilated with the wind variability. In our case, the instrumental precisions for the wind speed and direction measurements by the LiDAR are only of ±0.1 m/s and ±0.5°. Another source of uncertainty for the wind measurement is associated with the vertical interpolation of the wind profiles between successive measurement levels, but this error cannot be estimated. It has been shown in the literature (Yang et al., 2018; Morales et al., 2022) that low and unstable wind conditions could bring higher quantification errors for comparable mass balance method based on UAV measurements: a threshold of 2.3 m.s$^{-1}$ for minimum wind speed and 33.1° for maximum standard deviation of wind direction is considered as the limit for good wind conditions. We present the mean and standard deviation of the wind speed and the standard deviation of the wind direction at 1.5 m for each flight of the TADI campaigns in Tables S1 and S2. It can be noted that these variabilities are of several orders of magnitudes above the instrumental uncertainties of the wind measurements (±0.1 m/s and ±0.5°). In our case, only 26 flights would be considered under good wind conditions while 54 flights were performed under such bad wind conditions (see Table S1 and S2). Contrary to what was expected, no link has been noticed between the mean or standard deviation of the wind speed or the standard deviation of the wind direction and the relative errors of the quantifications (Figures S2, S3 and S4).

One would logically expect an improvement of precision when averaging the results of multiple quantification flights for the same source, as this would approximate a mean plume distribution and lower the effect of the turbulent nature of the wind, considering constant source and wind. However, no significant difference can be observed between quantifications of controlled release experiments based on 1, 2, 3 or 4 flights (Table 3 and Table 4): the dispersion of results is slightly lower for the quantifications based on 4 flights (minimum and maximum relative error of 19 and 26 % only), but this result cannot be considered statistically valid since it concerns only 3 controlled release experiments.

### 3.3.4 Discussion

Experiments conducted during the TADI campaigns allowed to validate our emissions quantification method, which depicted similar performances for CH$_4$ emissions on a wide spectrum of fluxes, ranging from order of magnitudes between $10^{-2}$ and $10^{+2}$ g.s$^{-1}$. The detection limit is most probably determined by the signal to noise ratio of the concentration measurements. As the flight plan distance to the source can be adapted for each flight to increase or decrease the level of signal (either to measure signal out the noise level, or to avoid saturation), absolute lower and upper detection limits depend from the conditions on the field (potential flight restrictions affecting the horizontal or vertical area covered by measurements, particular wind conditions, etc.).

If the validation of our method has been done specifically for CH$_4$ emissions quantification, it can easily be extrapolated to the quantification of CO$_2$ emissions, as long as the signal to noise ratios is sufficient (this can be controlled for each flight).

No clear link has been found between the quantification error and parameters like the mean and stability of the wind or the flow of emissions. One can suppose that the quantification error is determined either by multiple parameters simultaneously or by other parameters, among which some are difficult to quantify (e.g., distance of the transects to the source, angle between transects and wind direction, length of the transects, vertical gap between transects, oversampling or undersampling of the plume between different horizontal transects, missing top or bottom of the plume). One could expect uncertainties associated with the vertical interpolation to decrease while increasing the number of horizontal transects measured. This could be the subject of future experiments and analyses.

Analysing the sources of uncertainties in our quantification method, it is clear that a large source of uncertainties is linked to the knowledge of the wind speed and direction. Wind profiles measurement have been performed with a LiDAR during the TADI campaigns. As the emission sources were at low elevation from the ground and as the measurements were performed at a relatively short distance, thus with low vertical mixing, most flights were performed at low elevations, in particular in 2019 (typically below 12 m.a.g.l.). In such conditions, wind profiles measurements could have been performed with alternative devices such as multiple ultrasonic wind sensors sprayed along a vertical mast of a few meters, instead of a LiDAR which is unable to measure between the meteorological station at 1.5 m.a.g.l and the first LiDAR level at 11 m.a.g.l. In 2021, the distance between the flight plan and the source being generally longer than in 2019, some of the flights reached higher altitudes (up to 35 m.a.g.l.), thus requiring the use of a LiDAR. For low elevations (below the first LiDAR level) the uncertainties associated with wind speed measurements would be expected to be higher than for a within the range of levels monitored by the LiDAR. The interpolation of the wind speed between 1.5 m and 11 m was performed using a simple logarithmic regression, which does not necessarily perfectly match the real vertical wind profile. This brings a larger uncertainty and bias in our quantifications in the case of low plumes, such as those encountered during most of the TADI experiments. We expect better results at higher elevations within the elevation range of the LiDAR observations.Furthermore, the monitoring frequency of the employed Lidar technology was relatively low (15 to 20 seconds), which does not allow a good representation of the wind variability at the time scale of atmospheric turbulence at low elevation. In addition to the non-negligible distance between the LiDAR monitoring area and the UAV, this supports the need to develop high frequency wind monitoring directly on-board the UAV. In addition, conditions of low wind speed and variable wind directions were often encountered during the TADI campaigns, which is also challenging for emissions quantification as it is associated to more instabilities of the wind direction and thus an uneasy definition of the measurement plane. Considering these multiple suboptimal conditions, higher precisions could be expected for the monitoring of large and/or high sources such as offshore platforms, stacks or flares which rarely experience low wind conditions.

Our flux estimates from the TADI campaigns can be compared to the performances of other commonly used methods. As described earlier, our quantification method obtained 24 % of results between -20 and +20 % relative error compared to the true values, 80 % of results between -50 % and +100 % and all the results were within the range of -69 % to +150 % compared to the true values. Several technologies using UAVs, airplanes, or mobile ground measurements were tested and compared during the international Stanford/EDF Mobile Monitoring challenge (Ravikumar et al., 2019) at the Methane Emissions Technology Evaluation Center (METEC), in Colorado, US and at a facility near Sacramento, California, US. The performances of our method are better than

the those of the other techniques compared within this challenge: only one method (Seek Ops Inc., based on drone observations) had all quantifications between -90 % and 1000 %, but with only 36 % of quantifications between the -50 % to +100 % interval; while the best performance on the -50 % to +100 % interval was achieved by Ball Aerospace plane observations with 53 % of quantifications within this range. Emitted fluxes were generally lower for the Stanford/EDF challenge (from 0 to 0.1 g s$^{-1}$ on METEC and 0 to 7 g s$^{-1}$ at Sacramento) than for our TADI intercomparison experiments (from 0.01 to 150 g s$^{-1}$), but, as stated earlier, our results are similar on a subset of experiments focusing on the lowest emitted fluxes.

Other methods for $CO_2$ and $CH_4$ sources tracking and emissions quantification include measurements with CRDS analysers from cars. An evaluation of such technique coupled with an atmospheric inversion based on a Gaussian plume dispersion model has been carried out under conditions comparable with our study during a TADI intercomparison campaign in 2018 (Kumar et al., 2021). Results of this validation campaign depicted a good accuracy of the emissions quantification, with estimates of the $CH_4$ and $CO_2$ release rates with ~10 to 40 % average relative errors. But only a limited number of 16 out of 50 controlled releases could be monitored, as this technique is constrained by the ability to drive through the plume, which is not possible for high elevation plumes (in cases of high stacks or plume rise) or for wind direction incompatible with the road infrastructure.

A UAV-based $CH_4$ emissions quantification method with a near-field Gaussian plume inversion model (Shah et al., 2020) obtained large uncertainties compared to our method with respective lower and upper uncertainty bounds on average of 17 % ± 10 % (1σ) and 227 % ± 98 % (1σ) of the controlled emission flux. Gaussian approaches rely on hypotheses such as a well-mixed plume (problematic at a short distance from the source), a flat terrain, uniform and constant wind conditions, which are not necessarily true and may be less detrimental for mass balance approaches. The higher acquisition frequency of our analyser compared to this study is also a technical advantage which leads to better spatially resolved measurements and therefore an improved representation of the plume.

A recently published UAV-based emission quantification technique also relying on a mass bass approach (Morales et al., 2022), was tested on a short range of release rates (0.26 to 0.48 g.s$^{-1}$) and obtained average bias of -1 % and RMSE of errors of +69 %. These results are comparable with the average and standard deviation of our residuals (+7 % and +53 %), which supports the validity of the mass balance method for the quantification of greenhouse gases emissions. The main differences compared to our approach was the use of only low-level sonic anemometers to measure wind speed and direction, without a real monitoring of the vertical wind profile, and the quantification of $CH_4$ emissions exclusively with a heavier sensor (2.1 kg compared to 1.4 kg for our sensor).

### 3.4    Application to offshore oil and gas facilities emissions quantification

### 3.4.1    Protocol of offshore platforms monitoring campaigns

A one-day measurement campaign was conducted in the North Sea on April 2019 to quantify the emissions of two offshore platforms (hereafter named P1 and P2). These platforms are equipped with power generators and gas turbines driving the compressors, both emitting $CO_2$ to the atmosphere. Stacks of gas turbine are at 50 m above sea level (m.a.s.l.) with vertical ejection, stacks of power generator are at 30 m.a.s.l with horizontal ejection. The main source of $CH_4$ emissions is the gas venting system, at 80 m above the sea surface, emitting mainly methane with vertical ejection. Other potential minor sources of $CH_4$ are expected: fugitive emissions and unburnt $CH_4$ in the turbine smokes.

Measurements were carried out from a supply boat chartered on purpose by the company from Den Helder harbor, Netherlands (Figure 9,a.b.). The deck was used as a take-off and landing site for the UAV. The wind LiDAR installed on the deck recorded wind profiles at 10 levels between 15 and 300 m.a.s.l. (Figure 9,b). Real-time concentration measurements were visualized by a person assisting the pilot to adapt the UAV trajectory to the position of the plumes and manage wind direction fluctuations. The duration of each flight was of 10 to 20 minutes. Each flight can be assigned to a trial in terms of concentration recording and emission calculation. The first flight is often a detection flight aiming at localizing the plume and not always usable for emission quantification. Respectively 8 and 7 repeated flights and emissions quantifications were conducted for the monitoring of both platforms (see Table S3 in the Supplementary Materials).

Our UAV-based quantifications of $CO_2$ and $CH_4$ emissions are presented as relative differences to reference daily averaged emissions calculated by the operator of the platform. The reference emissions calculations from the platform are based on real measurements on the day of comparison, using mass balance and processing data (venting) and combustion balance (gas turbines and power generator). They are expected to be reliable for $CO_2$ emissions, as they are based on reliable input data (combustion flows, gas composition, $CO_2$ conversion of hydrocarbons), but assume the proper functioning of equipment, which can be a source of errors for $CH_4$ emissions(e.g. unexpected open valves or leaks). They also do not reflect the intermittency of the platform operation.

### 3.4.2 Offshore platforms emissions quantifications

During this campaign, the distances between the source and the measurement plane were varying between approximately 150 and 450 m depending on the flight (Supplementary Table S3). To match the vertical distribution of the plumes, originating from sources at typical elevations around 80 m.a.s.l., horizontal transects were performed within the range of 50 to 120 m.a.s.l. The signal to noise ratios obtained during the flights range from 78 to 4337 for $CH_4$ and from 66 to 523 for $CO_2$ (Supplementary Table S3), thus comparable for both species to the ratios obtained during the TADI campaign (for $CH_4$ only).

Figure 10 presents typical wind conditions for one flight (2_P2) of the offshore platforms emissions monitoring campaign. Stable wind directions were observed during this flight (Figure 10.a) with similar wind directions at for all horizontal transects. The absence of strong shear in the wind direction during our measurements allowed capturing emission plumes within a single measurement plane for each flight. The wind speed profile of this example is typical for this offshore campaign (Figure 10.b), with a logarithmic profile below 40 m and increasing wind speeds above this limit, typical for stable atmospheric conditions. The average and standard deviations of the wind speed and standard deviation of the wind directions at the elevation of the drone are presented for all flights in Supplementary Table S3. Similar stable wind conditions where encountered for all flights, with mean wind speeds ranging from $7.4 \pm 3.0$ to $10.6 \pm 2.9$ and standard deviations of the wind direction below 14.4°. These wind conditions were better than those encountered during the TADI campaigns.

Results of the emission quantifications of both offshore platforms are presented in Table 5. The quantified emission fluxes are presented in terms of relative difference compared to reference daily-average fluxes estimated for the platforms by mass balance and combustion calculation, thus non-representative of short-time variations of emissions.

For the quantification of $CH_4$ emissions, 13 flights were used mong 15 flights (7 for platform P1 and 6 for platform P2), the first flight for each platform being a short test flight to find the plume position. Mean $CH_4$ emission quantification for all 7 flights for P1 platform presents a 46 % relative difference compared to the reference vent stack expected emissions. This difference is of 12 % for the P2 platform. At the time scale of individual flights, large variations in the $CH_4$ emissions quantification are observed for both platforms, with estimates varying

between +8 % and +128 % for the P1 platform, and between -60 and +229 % for the P2 platform, compared to the daily reference emissions. For the P2 platform, the highest estimate of $CH_4$ emissions corresponds to a single flight (+229 %) largely above the average and standard deviations value of the other 5 flights (-31 ± 18 %). The vertical profile of $CH_4$ fluxes by transect levels for this particular flight (not shown) depicts an important flux of $CH_4$ at an elevation lower than the usual main plume observed for all other flights. It is therefore reasonable to interpret this

flux value as a short-time event of emissions from a different source than those used for the reference daily average estimates.

The mean values of the methane emissions quantification for all flights combined are comparable although higher than reference daily averaged emissions (+46 % and +12 % for P1 and P2). Our quantification method should be representative of the actual emissions of the whole platform, including fugitive emissions. Reference fluxes are

based on estimation using emission factors, gas composition and flow rate measurements or estimation. The higher methane emissions of platform P1 from our method compared to the reference emissions led to a review of some of the platform processes during which an unexpected emission was detected and repaired from a defect valve. A significant emission reduction is expected after the repair. Repeated measurements would be helpful to confirm the actual improvement.

Concerning the quantification of $CO_2$ emissions, 7 flights could be used for emissions quantification for platform P1, while only 4 flights are used for platform P2, as the $CO_2$ plume was not entirely captured by our flight plans during some flights, contrary to the $CH_4$ plume, as different sources are involved for both species. The $CO_2$ emissions quantifications are expressed as a relative difference to the daily averaged reference emissions. The estimated $CO_2$ emissions relative difference to the reference emissions are on average for all flights of -21 % for

platform P1 and -47 % for platform P2. Emissions quantifications of each independent flight provided variable results, with minimal and maximal values of -39 to +14 % for platform P1, and between -28 % and +2 % for platform P2, thus within the precision of our quantification method. Part of the temporal variability of the $CO_2$ emissions quantification of platform P1 could also be explained by the presence of a supply vessel which arrived and left the platform during the two flights with the highest emissions quantified. Part of the quantified $CO_2$

emissions of both flights could therefore be attributed to the emissions of this supply vessel. If only the other 5 flights are considered, the averaged quantification of $CO_2$ emissions would be of -31 % ± 18 % relative difference compared to the reference value for platform P1, with a maximum value of -20 %.

Altogether, our emissions quantifications depict large variations between the different flights, for $CO_2$ and more particularly for $CH_4$ fluxes (Table 5). Such variations can be linked with real short-term variations of the emissions

over the monitoring period, which are not reflected by the reference emissions values provided at a daily resolution only.

For some flights of this campaign, the measurements did not properly cover the entire plume cross-section vertically (values of $q(z)$ did not reach zero). Therefore, the plumes vertical boundaries were estimated from gaussian interpolations of the vertical distribution of $q(z)$. Better flight plans including measurements below and

above the plumes would be necessary for improved quantifications and will be an important requirement of future monitoring protocols.

## 4  Conclusions

This study presents a complete measurement system for the quantification of atmospheric emissions based on a new atmospheric $CO_2$ and $CH_4$ concentration analyser embarked under a UAV associated with a mass balance box

model.

In-lab validation of the analyser allowed to estimate its precision (10 ppb for $CH_4$ and 1 ppm for $CO_2$ when 615 averaged at 1 Hz) and depicted excellent linearity compared to a reference instrument, with good accuracy and repeatability (below 3% for $CH_4$ and 1% for $CO_2$). In-flight instrumental precision has been evaluated to 118 ppb in 2019 and 30 ppb in 2021 for $CH_4$ (depicting improvements of the analyser) and to 0.3 ppm for $CO_2$, whereas

for the in-flight accuracy, no intercomparison with a reference instrument was performed in-flight, which could be the subject of future work.

The controlled release campaigns on the TADI platform in 2019 and 2021 validated the complete emissions quantification method independently independently of the type of source or carrier, and showed better accuracy compared to other current top-of-the-art $CO_2$ and/or $CH_4$ emissions quantification techniques using either

multispectral camera, ground based CRDS (fix stations or mobile measurement in a car), wind and gas LiDAR, infrared camera including concentrations and emissions quantification system, or Tunable Diode LiDAR (Ravikumar et al., 2019; Shah et al., 2020; Kumar et al., 2021; Druart et al., 2021) and comparable performances for a similar technique also relying on UAV laser-based concentrations monitoring associated with a mass balance model (Morales et al., 2022). This measurement system was already applied on the field and extended in 2022 to

more than 100 oil and gas facilities, offshore and onshore, from tropical to high latitude environments, which will be the subject of upcoming publications. It has a wide range of potential applications, for the quantification of $CO_2$ or $CH_4$ sources of diverse anthropogenic or natural origins, such biogas plants and landfills.

Field applications of our measurement system to offshore oil and gas platforms revealed several assets compared to similar quantification campaigns previously conducted from aircraft or boats. Compared to aircraft-based

monitoring, UAVs have the advantage to fly below 300 m high and close to the facilities (distance around 250 m from offshore platforms), allowing the monitoring of the entire plume and the identification of the main sources. The real time monitoring of the concentration on the ground, associated with the high speed and reactivity of the UAV, provides the possibility for the pilots to adapt the trajectory and fly within the plume despite its meandering. The UAV high speed also allows monitoring of an entire plume within a few minutes, thus representative of a

quasi-stationary state, preventing for example double measurements of the same plume when it is meandering, which could occur with measurements conducted from a low-speed vessel. The high frequency of observations (conducted at 24 Hz for these campaigns) allows us to apply a mass balance quantification method which does not require a 2D interpolation (e.g. Kriging) of the measured concentration data, but only an interpolation along the vertical direction.

Nevertheless, our measurement system would benefit from further improvements, among which improved performances of the instrument, of the monitoring protocol or of the modelling.

The simple mass balance method presented here proved able to provide emissions quantifications at low computing costs. Uncertainties were only empirically determined in this study. The quantification method needs further development to propagate the different sources of uncertainties and provide an uncertainty of the emissions quantification. Other modelling approaches could be used: this mass balance method requires concentration measurements throughout an entire plume cross-section, which is not always possible to perform on the field, due to restrictions of the UAV area of operations caused by obstacles or prohibited flight zones. An inverse atmospheric modelling approach with atmospheric dispersion models, will also be tested in the future (e.g. Darynova et al., 2023). Further experiments and analyses are also required to determine the limits of this method related to the wind stability and minimal wind speed.

A more precise recording of the horizontal and vertical UAV positioning has already been introduced with the use of RTK GPS positioning, facilitating data post-treatment. Future technical development of our method will include wind speed measurements directly on-board the UAV, replacing the LiDAR wind profile measurements for an easier and more cost-effective field deployment. This should also improve the wind speed measurements at elevations below the lowest level of wind LiDAR measurements (typically below 10 m.a.g.l.). A fully automatized UAV operation is also being developed, with UAV track adapting to the plume position, aiming at regular quantifications of O&G facilities.

Future improvements will be made to our greenhouse gases sensor. CO concentrations measurements are being developed for future versions of our instrument, which will allow the calculation of a complete combustion efficiency balance for various types of sources of the O&G sector, such as flares. Further weight reduction and adaptations of the instrument will allow it to be embarked by a larger spectrum of air carriers, including VTOLs (Vertical Take-Off and Landing) UAVs, which have a longer autonomy and fly and higher speed. This will open new applications to monitor emissions of larger scale sources such as larger industrial facilities, natural sources or small cities.

**Author contribution**

The manuscript writing has been initiated by J.L.B., completed by N.G., L.D. and L.J. and corrected by all co-authors. The AUSEA sensors and associated spectrometric inversion algorithms have been developed at the GSMA by L.J., J.C., T.D., J.B., N.C., N.D., G.A., F.P. In-lab validation of the AUSEA sensor has been performed by J.L.B. and D.C. Mass balance emission quantification programming and computation of emissions quantifications were conducted by L.D. and N.G. Field monitoring campaigns were conducted by A.M, O.V. and L.D.  The project has been initiated and coordinated by L.J. and L.D., O.D., C.J., M.F.B.

**Competing interests**

The authors declare that they have no conflict of interest.

**Acknowledgments**

The instrumental development and the quantification algorithm were funded via a collaboration between the GSMA laboratory of the Université de Reims Champagne-Ardenne (URCA) and the R&D LQA laboratory of TotalEnergies, within the frame of the AUSEA project and the common laboratory LabCom LYNNA (hosted by CNRS, URCA and TotalEnergies).

Observations around offshore platforms were conducted from a supply boat chartered by TEPNL from Den Helder
harbor. We thank the crew for their assistance and all the affiliated people that helped us to manage those experiments.

Participations to the TADI campaign were possible thanks to the facilities and assistance of the TotalEnergies TADI teams from the PERL (Pôle d'Etude et de Recherches de Lacq), who organised the controlled releases.

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

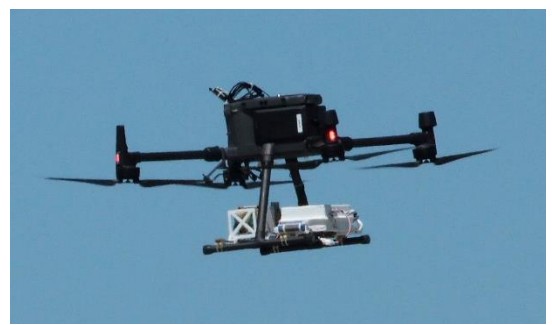

**Figure 1. Picture of the AUSEA 112 analyzer mounted on a DJI M300.**

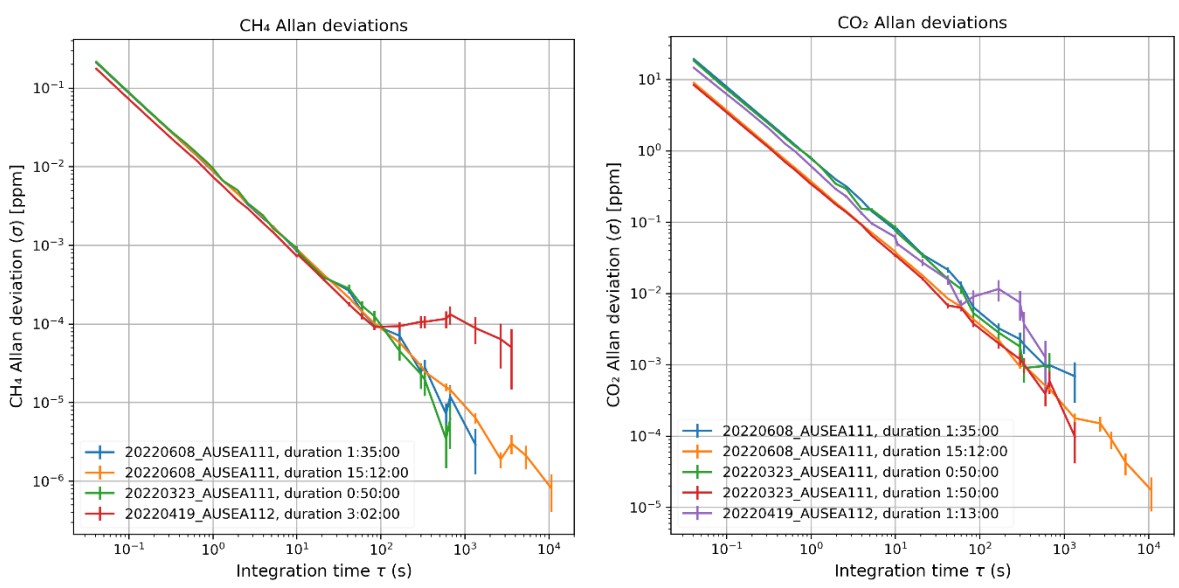

**Figure 2 Allan deviations calculated for multiple stability experiments with analysers AUSEA_111 and AUSEA_112.**

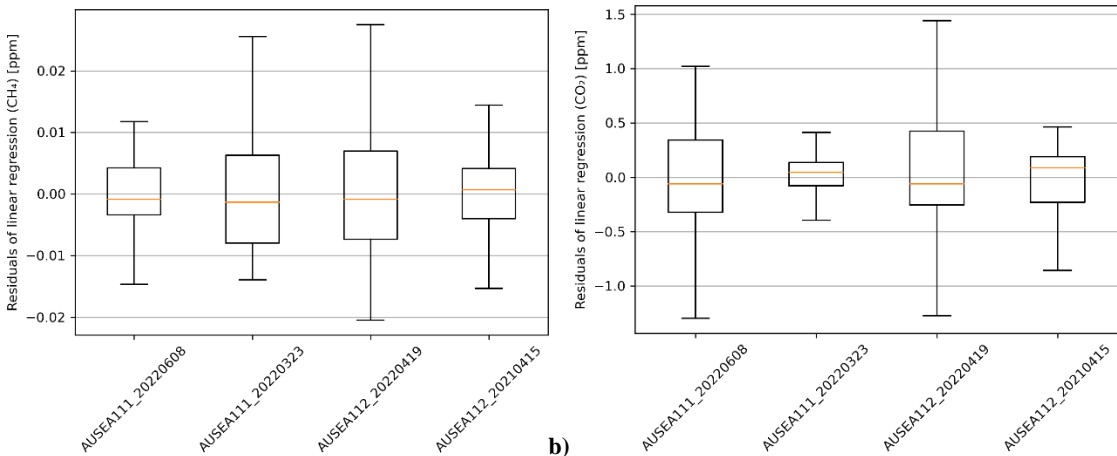

**Figure 3. Boxplots of the residuals of the linear regressions for each linearity experiment of the AUSEA sensor against a reference Picarro CRDS analyser in a temperature-controlled environment at a 5 s temporal resolution. Boxplots depict the first and last quartile (lower and upper borders of the boxes), median (orange line) and minima and maxima (lower and upper ticks, defined as the first and last quartile plus or minus 1.5 times the interquartile range), without outliers.**

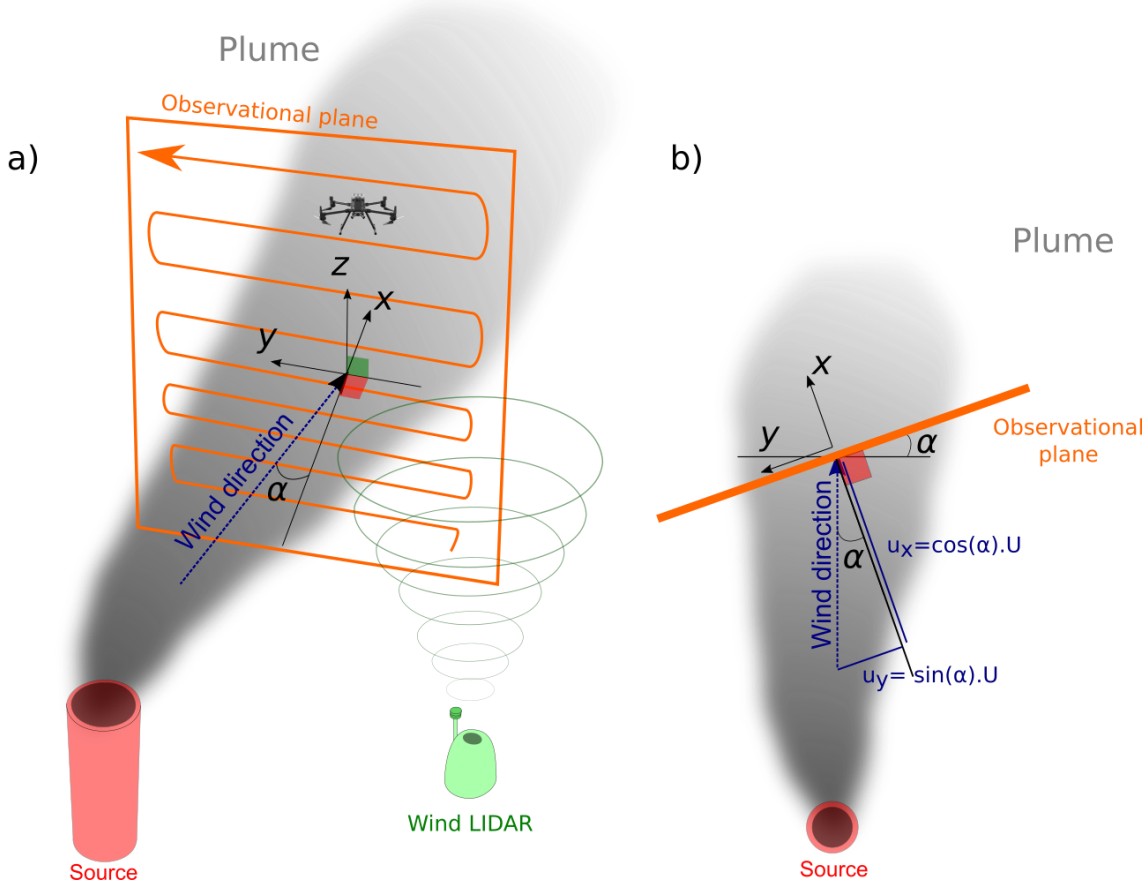

**Figure 4. Schematic representation of the observation protocol: general 3D view (a) and top view (b). The source (in red) emits a plume (grey shade). The UAV monitors the concentrations along a flight path (orange arrow), constituted of multiple horizontal transects, within a vertical observational plane, represented by the orange quadrangle in (a) and the orange line in (b). The angle between the orthogonal to the observational plane and the wind direction is noted $\alpha$. A wind LiDAR (green) measures the wind direction and speed at several elevations.**

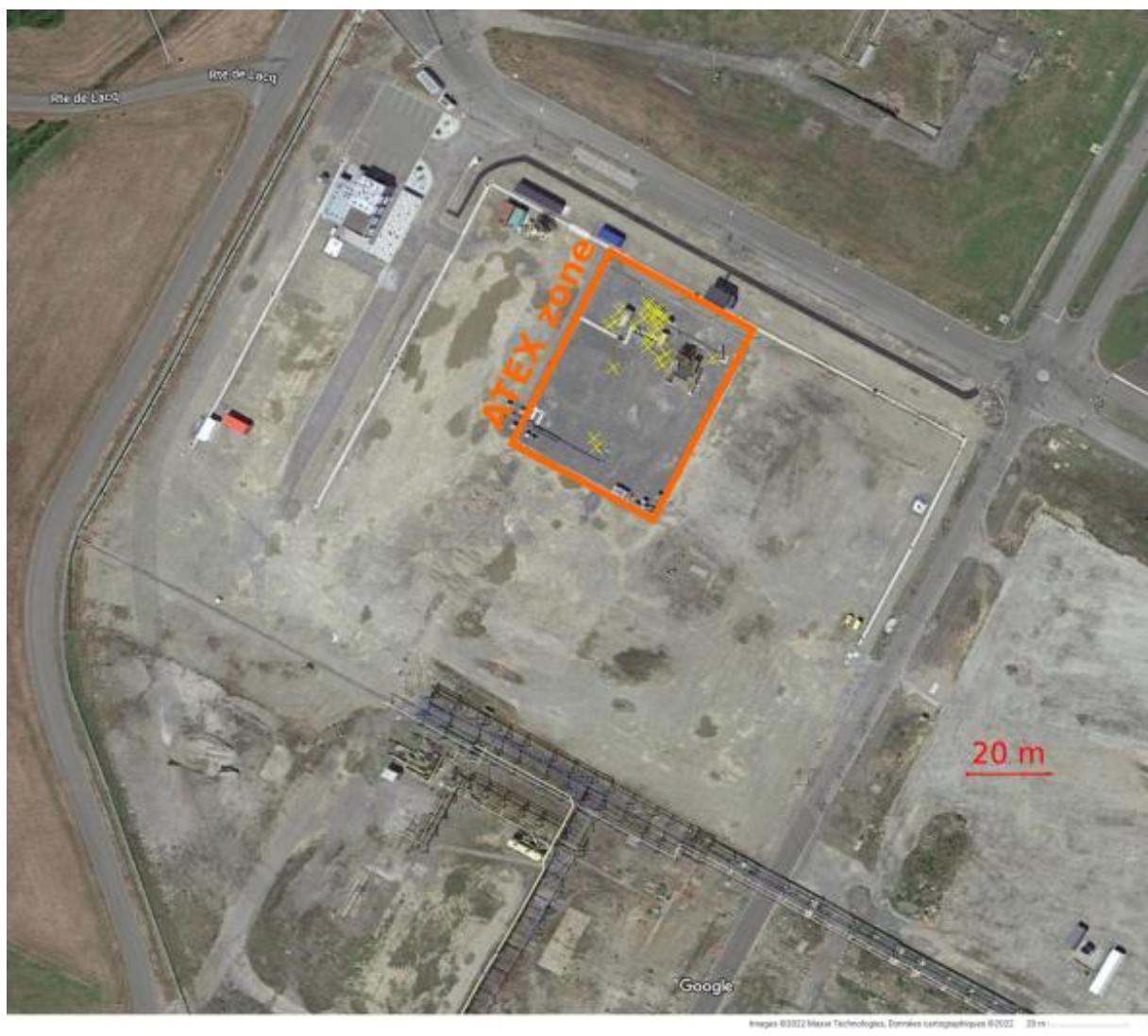

**Figure 5. Aerial view of the TADI platform with location of the emission sources (yellow crosses) and the Explosive Atmosphere area (ATEX zone, depicted as an orange square). Maps Data: Google, ©2022 Maxar Technologies.**

a)

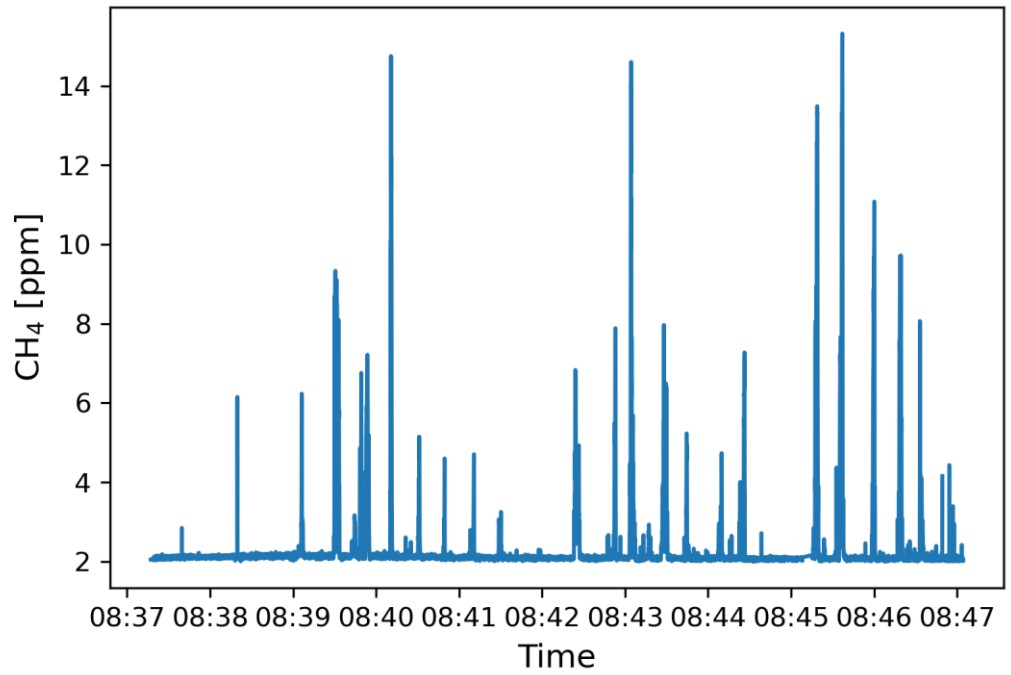

b)

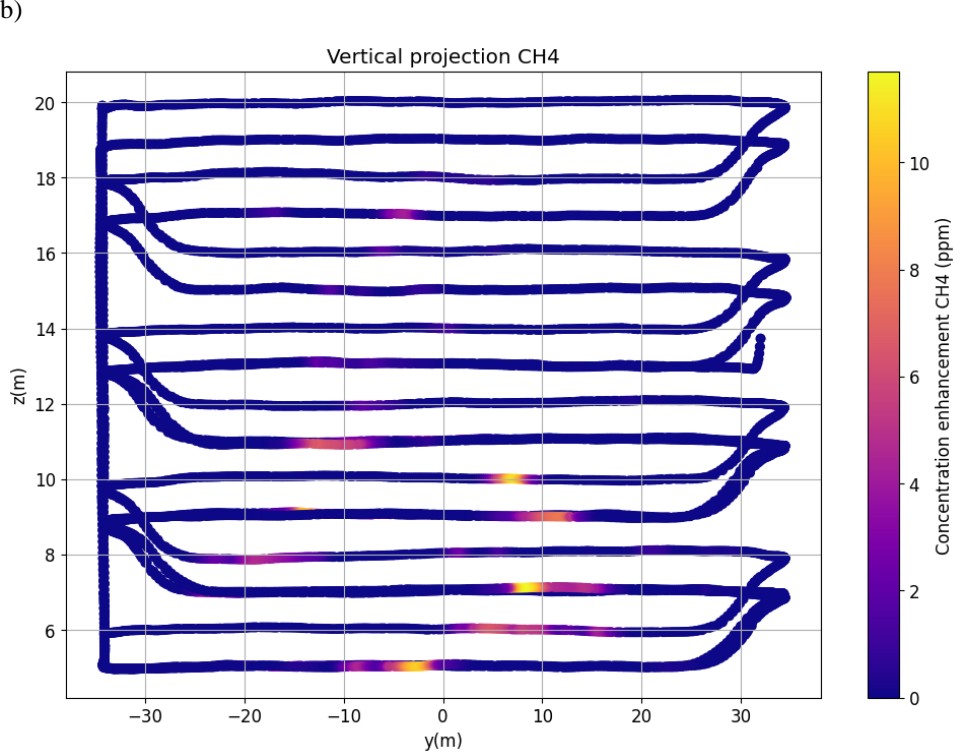

**Figure 6. Measured CH₄ concentrations (ppm) during a flight of TADI 2021 campaign corresponding to the monitoring of the 2021W36/10-22 controlled release experiment performed on 2021-09-10: (a) time series and (b) distribution along the y horizontal axis (in meters) and the vertical axis (altitudes in m.a.g.l.) of the CH₄ concentrations above the background level.**

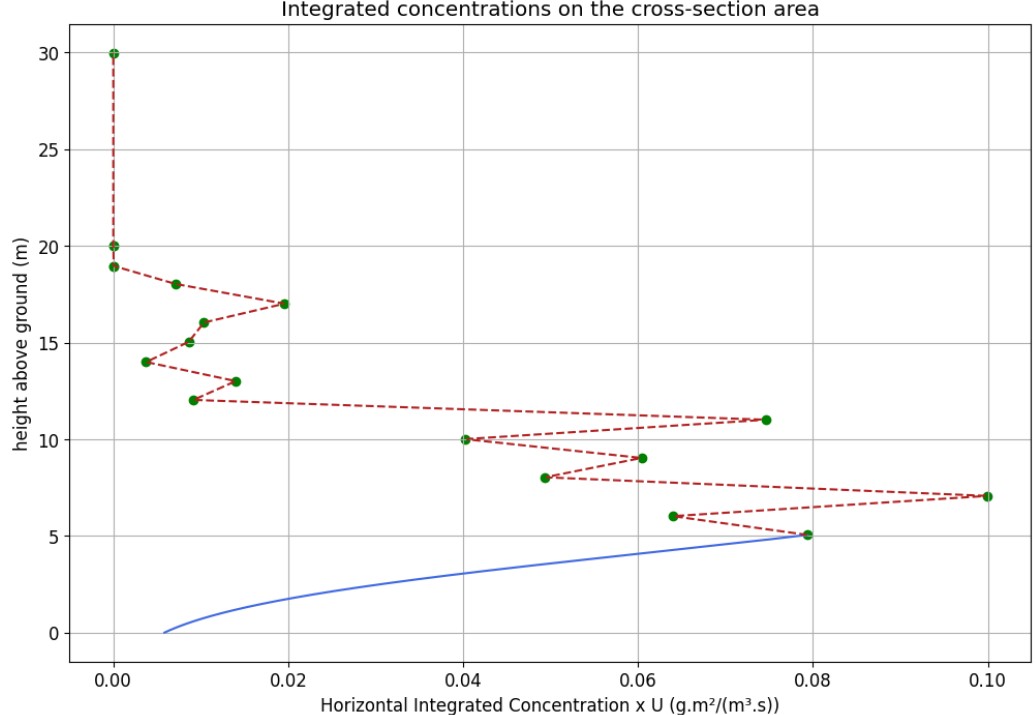

**Figure 7. Vertical distribution of the horizontal flux components q(z) (in g.s$^{-1}$.m$^{-1}$) at each transect level *z* (green dots), along with the interpolation used for the vertical integration (red-dotted curves for linear interpolation and plain blue curve for logarithmic interpolation), for the same flight as Figure 6.**

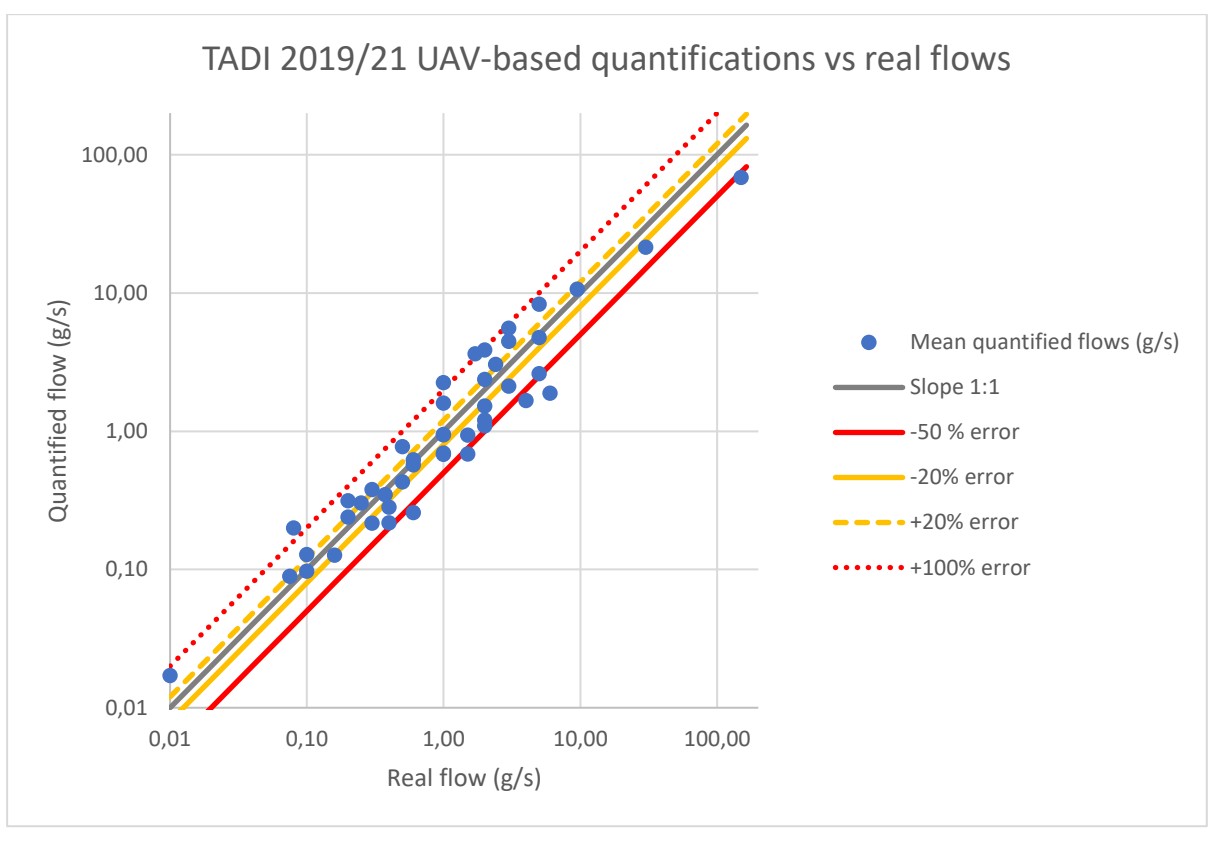

**Figure 8. Comparison of emissions quantifications as a function of the real CH$_4$ emissions fluxes (in g/s), as blue dots. A log-log scale is used. The plain grey line indicates the 1/1 slope or 0 % error. Plain and dashed yellow lines respectively**

**indicate the -20 % and +20 % relative errors limits. Plain and dotted red lines respectively indicate the -50 % and +100 % error limits.**

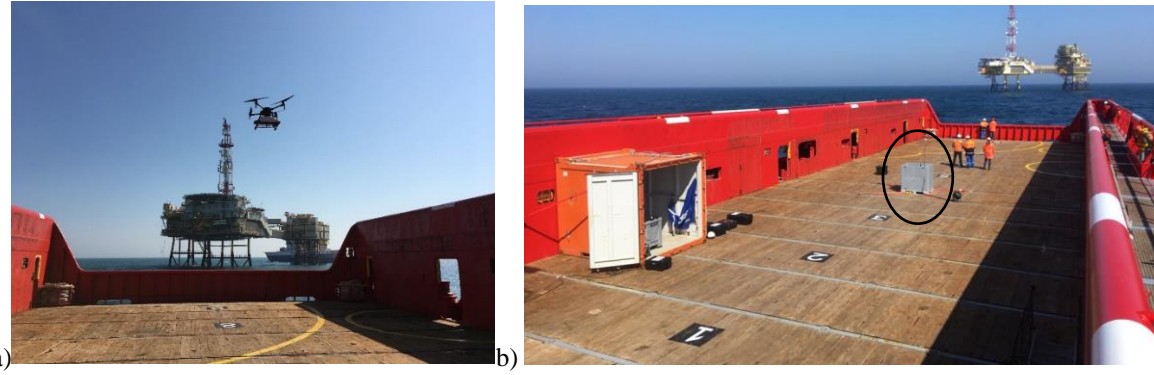

a)                                        b)

**Figure 9 : Picture of operations nearby offshore platforms in the North Sea on 2019-04-19, showing (a) the UAV equipped with the AUSEA sensor and (b) the deck of the supply vessel serving as take-off and landing site for the UAV, with the wind LiDAR (black circle).**

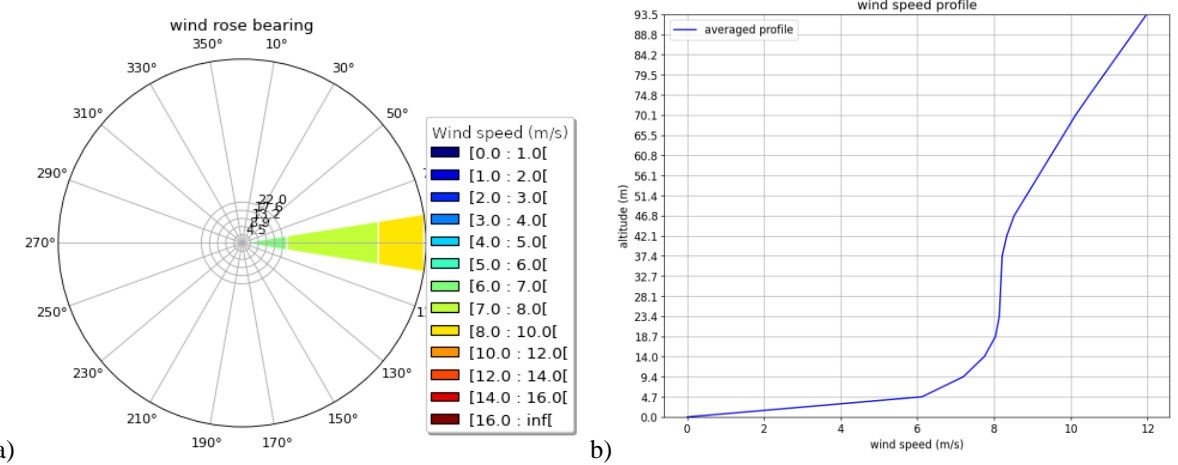

a)                                                             b)

**Figure 10. Typical weather conditions for flight 2_P2 of the offshore platform monitoring campaign: a) Distributions of wind directions (percentage, with 20° resolution) for different wind speeds classes (color scale, in m/s) during flight measured by the LiDAR at the lowest level. b) Averaged wind speed (m/s) vertical profile (in m.a.s.l) over the flight duration, measured by wind LiDAR at each of its monitoring level.**

| Instrument | Date | Duration | Species (unit) | $\sigma_{0.5\ s}$ | $\sigma_{1\ s}$ | $\sigma_{10\ s}$ | $\sigma_{60s}$ |
|---|---|---|---|---|---|---|---|
| AUSEA111 | 2022-06-08 | 1 h 35 min | $CH_4$ (ppb) | 18 | 9 | 0.8 | 0.1 |
| AUSEA111 | 2022-06-08 | 15 h 12 min | | 18 | 9 | 0.9 | 0.1 |
| AUSEA111 | 2022-03-23 | 50 min | | 19 | 10 | 0.9 | 0.2 |
| AUSEA112 | 2022-04-19 | 3h 2 min | | 15 | 8 | 0.7 | 0.1 |
| AUSEA111 | 2022-06-08 | 1 h 35 min | $CO_2$ (ppm) | 1.6 | 0.8 | 0.1 | 0.01 |
| AUSEA111 | 2022-06-08 | 15 h 12 min | | 0.8 | 0.4 | 0.04 | 0.01 |
| AUSEA111 | 2022-03-23 | 50 min | | 1.5 | 0.8 | 0.1 | 0.01 |
| AUSEA111 | 2022-03-23 | 1 h 50 min | | 0.7 | 0.3 | 0.03 | 0.01 |
| AUSEA112 | 2022-04-19 | 1h 13 min | | 1.3 | 0.6 | 0.06 | 0.01 |

**Table 1 : Precisions of the AUSEA111 and AUSEA112 analyzers at given frequencies (0.5 s, 1 s, 10 s, 60 s) derived from**
**Allan deviations of stability experiments performed at different dates, expressed in ppb for CH₄ and ppm for CO₂.**

| Instrument | Date | Species | Minimum (ppm) | Maximum (ppm) | Slope | Intercept (ppm) | $R^2$ | N |
|---|---|---|---|---|---|---|---|---|
| AUSEA112 | 2021-04-15 | $CH_4$ | 2.1 | 20.0 | 1.009 | +0.03 | 1.0 | 30751 |
| | 2022-04-19 | | 2.2 | 20.0 | 1.032 | +0.163 | 1.0 | 17169 |
| AUSEA111 | 2022-03-23 | | 3.2 | 20.0 | 1.008 | -0.018 | 1.0 | 16267 |
| | 2022-06-08 | | 2.2 | 20.0 | 0.991 | +0.024 | 1.0 | 5943 |
| AUSEA112 | 2021-04-15 | $CO_2$ | 429.0 | 509.9 | 1.007 | +6.538 | 1.0 | 30751 |
| | 2022-04-19 | | 454.9 | 646.3 | 1.009 | +6.2 | 1.0 | 17169 |
| AUSEA111 | 2022-03-23 | | 465.3 | 657.8 | 1.011 | +10.702 | 1.0 | 16267 |
| | 2022-06-08 | | 532.3 | 861.4 | 0.995 | +17.7656 | 1.0 | 5943 |

**Table 2 : Results of the linearity experiments for instruments AUSEA111 and AUSEA112 performed at different dates, for the $CH_4$ and $CO_2$ measurements: range of concentrations covered by the experiments (minimum and maximum values), slope and intercept of the linear regressions of the distributions (only values below 20.0 ppm for $CH_4$) and associated $R^2$ values and number of data points at a 5 seconds frequency resolution used for the linear regression.**

| Median | Average | σ | Minimum | Maximum | Number of experiments |
|---|---|---|---|---|---|
| **Quantifications relative errors (%)** | | | | | **Number of experiments** |
| **With 1 quantification flight** | | | | | |
| -28 | -8 | 53 | -58 | 125 | 19 |
| **With 2 quantifications flights** | | | | | |
| 12 | 17 | 57 | -69 | 149 | 19 |
| **With 3 quantifications flights** | | | | | |
| 27 | 23 | 49 | -32 | 70 | 4 |
| **With 4 quantifications flights** | | | | | |
| 19 | 21 | 4 | 19 | 26 | 3 |
| **All quantifications** | | | | | |
| **-5** | **7** | **53** | **-69** | **149** | **45** |

**Table 3. Statistics of the relative errors of quantifications during the TADI campaigns, for the release experiments quantified by 1, 2, 3 or 4 independent flights and for the total of all quantifications.**

| | | | Real emitted fluxes categories (g/s) | | | | | |
|---|---|---|---|---|---|---|---|---|
| | | | [0,01-0,3[ | [0,3-1[ | [1-2[ | [2-5[ | [5-151[ | [0,01-151] |
| | | Average relative error (%) | 38 | -10 | 15 | 7 | -18 | |
| **Relative error categories** | [-20% : +20%[ | **Number of experiments** | 3 | 3 | 2 | 1 | 2 | 11 |
| | [-50% : +100%[ | | 8 | 8 | 6 | 9 | 5 | 36 |
| | [-69% : +150%[ | | 9 | 10 | 9 | 10 | 7 | 45 |

**Table 4. Statistics of the quantifications results for different categories of real emitted fluxes: between 0.01 and 0.3 g/s, between 0.3 and 1 g/s, between 1 and 2 g/s, between 2 and 5 g/s, between 5 and 151 g/s and for the whole range between 0.01 and 151 g/s. The average relative error is given in % for each category, as well as the number of controlled release experiments for which the quantifications reached relative error categories between -20 and +20 %, between -50 and +100 % and between -69 % and +150 %. Underlined numbers correspond to the total number of controlled release experiment within each real emitted flux category.**

| | Relative errors to site calculations (%) | | | |
|---|---|---|---|---|
| Species | $CO_2$ | | $CH_4$ | |
| Platform (number of flights) | P1 (7 flights) | P2 (4 flights) | P1 (7 flights) | P2 (6 flights) |
| Minimum | -39 | -28 | 8 | -60 |
| 1st Quartile | -36 | -26 | 28 | -33 |
| Median | -23 | -19 | 32 | -21 |
| 3rd Quartile | -12 | -9 | 51 | -17 |
| Maximum | 14 | 2 | 128 | 229 |
| **Mean** | **-21** | **-16** | **46** | **12** |

**Table 5. Statistics of the distribution of quantified emissions for all 7 flights associated to each platform P1 and P2, for $CO_2$ and $CH_4$, expressed as relative differences (in %) to the reference daily average emission rates obtained by mass balance and combustion efficiency calculations.**