# Peer review of "A measurement system for CO2 and CH4 emissions quantification of industrial sites using a new in situ concentration sensor operated on-board Unmmanned Aircraft Vehicles"

_Atmospheric Measurement Techniques, 2022_

## Author Response (AR1)

Our answers to the referees are presented in **bold** characters.

- **RC1**: 'Comment on amt-2022-334', Anonymous Referee #1, 03 Jun 2023

Bonne et al., presented a new light-weight open path laser absorption spectrometer for in situ $CO_2$ and $CH_4$ concentration measurements, and applied the analyzer in the fields to quantify the emission rates of a series of controlled releases of $CH_4$ and $CO_2$ at the TotalEnergies TADI test platform at Lacq, France and of two offshore oil and gas platforms in the North Sea. Gas LiDAR was used to obtain vertical profiles of wind speeds and directions on site, which were used in the mass balance approach to calculate the emission rates. The paper is well structured, and both the laboratory and field experiments results could be useful to further develop the application of quantification of greenhouse gas emissions using UAVs. On the other hand, the paper would have benefited from more stringent discussion of the uncertainties of the measurements and the quantification using the mass balance approach. The paper is suitable for publication after addressing my following concerns.

General comments:

1. It would be very helpful for the readers to see some real measurements from the instrument, e.g., time series of measurements, and concentrations along the measurement plane. Note that the precision and linearity were derived from laboratory experiments under stable temperatures, which may not represent the real conditions in the field. Do the authors have an idea about the stability or the accuracy of the concentration measurements in the field? Is there any field comparison data?

**A figure has been added (Figure 6) showing an example of $CH_4$ concentration measurements during a flight of the TADI campaign.**

**Concerning the stability of the concentration measurements in the field, it indeed differs from the laboratory measurements, as the perturbations linked with the UAV vibrations and air circulation increase the noise level compared to the tests in-lab. More details have been provided about the in-flight precision of the instrument: to estimate the noise level of each flight, we consider the median value of the rolling 1 second standard deviation.**

**Concerning the in-flight accuracy of the instrument, no experiment has been performed allowing an intercomparison with a reference instrument. This could be the subject of future work.**

2. What does the analyzer measure? Note that greenhouse gas concentrations are reported as dry mole fractions. How were the water vapor effects accounted for? I understand that the analyzer also measures H2O, however, which was not shown or discussed.

**Our analyzer does not perform any spectral measurement of H2O. However, an InterMet iMet-4 radiosonde has been integrated to the analyzer with an acquisition frequency of 1Hz, which does record temperature (repeatability of 0.2°C, response time at 2s for still air and <1 s at 5 m.s-1 and 1000hPa) and relative humidity with response time of 0.6s at 25°C (5.2s at 5°C) and repeatability of 5%. The humidity is**

**considered for the calculation of the concentration based on the absorption spectra. The dilution effect of the water vapor is considered for the mass balance calculations.**

3. For the mass balance approach, even though the measurement frequency is rather high, interpolation of measurements along the vertical direction was needed. How was the interpolation done? I guess it was done in a similar way as Kriging. Again, it would be very helpful if some figures can be shown.

**We don't interpolate the greenhouse gases concentrations vertically: we first calculate the integral along the y axis of (concentration above background x wind) for each horizontal transect. This provides a linear flux for each level (in $g.s^{-1}.m^{-1}$). We linearly interpolate these values between the different vertical levels to calculate the vertical integration which provides the complete flux. For the upper and lower levels, if the values do not reach zero, we fit the distribution with a gaussian curve to complete the missing values.**
**Therefore, there is no Kriging of the concentration values in our calculation.**
**The description has been modified to describe these different steps more precisely. An example figure has been added (Figure 7) to illustrate the vertical interpolation of all horizontally integration values along each transect.**

4. The addition of a LiDAR during the measurements is certainly appreciated. However, the uncertainty of LiDAR measurements was not sufficiently discussed, especially when most flights were performed at low altitudes. Are LiDAR wind measurements at low altitudes reliable? What are the associated uncertainties?

**The LiDAR measurements are certified by the manufacturer and reliable (controlled against tall tower equipped with anemometers, precision of 0.1 $m.s^{-1}$ and 0.5°) from 10 to 200 m above the LiDAR window (typically 1 m above the ground). Complementary measurements at 1.5 m above the ground are performed by a sonic anemometer.**

**The largest source of uncertainty is therefore the interpolation between the monitoring levels of the LiDAR. For the values between 1.5 and 11 m, interpolation was performed with a very simple logarithmic function matching the values at 1.5 m and at 11 m. This might be a source of errors for our method. Associated uncertainties are however hard to estimate. This has been added in the revised manuscript.**

**Uncertainties linked with the LiDAR measurements are also linked with the low frequency of the measurements (15 to 20s), which does not allow to represent the turbulences of the low atmosphere.**

**This was already discussed in the article on the TADI results, but this discussion has been extended.**

Some detailed comments:

L11-37: the abstract can be written more concisely.

**The abstract has been reworked and is now more concise.**

L145: still, some more brief description of the system will be appreciated, e.g., how temperature and pressure variations are taken into account. How is water vapor used? How is the spectra data used to derive the mole fractions?

**More details have been added about the instrument: there is no spectroscopic measurement of the water vapor but the humidity is measured by the iMet-4 sensor. The effect of water vapor is considered in the theoretical spectral absorption model used to derive concentrations from the absorption spectra. The humidity is also used to calculate dry basis concentrations, considering the dilution effect of water vapor.**

**We believe the technical details on the spectroscopy are not the subject of the publication and have been already detailed in the papers presenting the original AMULSE instrument (Joly et al. 2016, 2020) from which the AUSEA instruments were adapted, keeping the same spectroscopic principles.**

L246: Similar flight protocols have been presented earlier, e.g., in Andersen et al., 2021, and Morles et al., 2022. Also, a minimum wind speed was used to select appropriate flights for analyses. How was the wind speed considered in this paper?

**More details about the wind conditions of all flight have been added in this revised manuscript. Concerning the validation experiments during the TADI 2019 and 2021 campaigns, no direct link has been found between the mean wind speed, the wind speed variability or the wind direction variability, and the relative errors of the quantifications. During the TADI 2019 and 2021 campaigns, many flights were performed under low or unstable wind conditions (under the classification of Morales et al. 2022). However, no significant effect on the error quantification has been found. It is possible that other variable parameters between the different flights also affect the uncertainty of the method and partially hide the link between wind speed and quantification error.**

L249: under the wind should be downwind

**This has been corrected.**

L270: what is the unit of Cp? Note that the water vapor effect needs to be considered, as is in eq. (4) in Andersen et al., 2021.

**The concentrations are expressed in $g.m^{-3}$. We consider the effect of dilution on water vapour and use dry basis concentrations of water.**

L337: plan should be plane

**This has been corrected.**

Reference:

Andersen, T., Vinkovic, K., Vries, M.d., Kers,B., Necki, J., Swolkien, J., Roiger, A., Peters, W., Chen, H.:Quantifying methane emissions from coal mining ventilation shafts using an unmanned aerial vehicle (UAV)-based active AirCore system, Atmos. Environ.: X, Volume 12, 100135, ISSN 2590-1621, https://doi.org/10.1016/j.aeaoa.2021.100135, 2021.

- **RC2**: 'Comment on amt-2022-334', David Noone, 19 Jun 2023

Major points

The paper outlines an extension of an earlier instrument development, to now simultaneously obtain methane and CO2 concentrations. The reported precision and stability over ambient measurement ranges are impressive, and able to provide critical information for a variety of emission and monitoring tasks. The performance of the instrument is impressive, but accomplishing this with the weight, power and size limits marks the limits of current best practices. The paper findings focus on comparison with other instrumentation during a field experiment.

The description of the new instrumentation is quite brief, and could be strengthened with some more detail to satisfy the reader of the additional performance relative to earlier work. The lack of detail doesn't fully document the technical aspects of the instrument to a degree that paper could stand on its own as a work describing instrument development. On the other hand, the presentation of a series of field based results is a sound demonstration of the strategy as a whole and is a useful pairing to the instrumentation overview.

**The technical details of the instrument development have been provided in earlier publications presenting the original AMULSE instrument (Joly et al. 2016, 2020), which is the core of the instrument and has not been modified. More details are given about the changes which have been made to adapt it to the UAV carrier and the telemetry. Providing more details on the spectroscopy is not our aim for this paper.**

The single largest limitation in the paper, however, is the lack of an error budget in the flux estimates. There is a description of the instrument precision, but there is no attempt to account for errors in the flux estimate. Issues on homogeneity of wind, plume structure, plume sampling, temporal variability, variation in the background and limitations in the assumptions leading to a mass balance approach ("model error") are all missing. In fact, the direct propagation of measurement precision on the final flux estimate was not given. This needs to be remedied. The revised paper should described an approach (and ideally a full account) of errors in the estimation method. One advantage the authors have is that they have occasion to have knowledge of the actual emission during controlled release, which allows a check not only on the quoted means (as is done), but some assessment of those means relative to the expected error.

I have a hunch that once done, the precision of the instrument will not be the leading source of uncertainty. As such, the authors can claim that their instrument meets the measurement need for these types of problems. However, unless the error budget is done, one can not place any confidence in the fact the instrument or methods are robust enough.

**More details have been provided concerning the evaluation of different sources of uncertainties: wind and concentration measurement uncertainties, signal to noise ratio of the concentrations, wind variability. Evaluating the impact of these uncertainties on the errors of quantification did not reveal much link between the variability of these parameters and the errors of quantification, apart from a limited effect of the signal to noise ratio. It seems that most of the uncertainty of the final method is either linked with a combination of these effects, and/or to the effect of hidden parameters which are more difficult to quantify (undersampling or oversampling of the plume with variable gaps**

**between transects, different durations to perform the full scan of the plume depending of the size of the plume, distance of the source and monitoring flight plane, extrapolation of top/bottom of the plumes if not monitored, angle between transects and wind directions, uncertainty associated with vertical interpolation, etc.).**

**A direct propagation of the uncertainties is however still missing in our algorithm, and will be the subject of future development. But based on the evaluation of the different sources of uncertainties which are possible to quantify, it is possible that the direct propagation of uncertainties finally does not match the errors of quantification.**

**Therefore, we chose to evaluate the uncertainties empirically, based on the results of the controlled release experiments of the TADI campaigns, which is representative of a global uncertainty on the field, knowing that it is a worst case scenario (low and unstable wind conditions, low elevated plumes and UAV flight elevation below the LiDAR monitoring levels) and the most of the real world field measurements are performed under better conditions (plumes at higher elevation with stronger wind and LiDAR monitoring levels matching the UAV elevation).**

The paper presets results of source flux estimates but does not describe if these are useful. Are the estimates sufficiently robust (small enough error) to be of practical use? A discussion section (and possibly a motivating section in the introduction) that lays out the measurement need in terms of total error (perhaps a percentage or perhaps a total value) is needed.

During revision, the authors should look for clarity in what they wish to use as a title and what to emphasize in the abstract. At the outset, the paper has focus on instrumentation, but in the end claims the focus is on flux estimation technique, while the bulk of the results focus on case study applications. I caution that the lack of specific development of the estimation technique make the claim that there is method development difficult to argue. The bulk flux estimate from a mass balance method was not developed, and the error budget not described. While I do suggest the error budget be added during revision, it is also wise to ensure the text focuses toward the comparative natured of the results.

To this end care is needed as, for instance in the conclusion (L497-501) appears to suggest the technique itself was devised here. This is overstating the matter. Re-phrase sections like this to highlight that it is the combination of the sensor, the UAV platform, and the application of a mass balance method when used together that represent a measurement system. The essential new work done here, to my mind, is in all three areas: using a newly developed sensor designed for UAV work, using data from the sensor to drive a mass balance calculation (albeit, without error characterization), and then testing it in pragmatic cases that illustrate possible wide application.

**These suggestions have been taken into account for the revision of the manuscript.**

Minor points

L24. TADI should be spelled out in the abstract.

**This has been modified.**

L75. "atmospheric concentration measurements can be operated´ should be instruments that are operated. (measurements can be obtained by remotely operated instruments, …etc)

**Thank you for this language correction. Done.**

L84 aircrafts - > aircraft (no "s"). Also, perhaps surface platforms rather than slow. Cars and ships can move quickly, and more quickly that some aircraft. Clarify your point, does speed matter, is the type of platform the key here.

**Ok, corrected.**

L88: Check phrasing "solutions …. To types of solutions" (you men "challenges?"). Also you might say UAV plat forms hold potential for these characteristics. It is not clear that they are currently cost saving, for instance.

**Ok, corrected.**

L89: You make the case for boats, but I suspect you excluded autonomous boats, which would have some similar benefits as autonomous aircraft.

**Yes, in terms of speed and distance to the source for example, but not it terms of vertical elevation.**

L105. Check phrasing. The "L" in DIAL is LIDAR: differential absorption LIDAR.

**Indeed. This has been corrected.**

L109: What does "high quality" mean in this context?

**Good remark, this actually depends from the expected signal to noise ratio and thus from the context. In the context of oil and gas industries, high range of concentrations are measured, so there is a need for high sensitivity range but the precision can be lower than for low and diffuse sources.**

L128: It would be useful to also not some of the limitations of open path measurement here. Close path DAS approaches have advantages as well.

**Indeed, close path DAS are less sensitive to external perturbations such as temperature or pressure changes, or external light sources. This has been added to the manuscript.**

L 155: embarks is the wrong word.

**"embarks" was replaced by "is equipped with"**

L170/171: exploited is the wrong word

**"Exploited" was replaced by "analysed"**

L180: In this section, it would be useful to provide some comparative values from other published work. Perhaps quote precision from commercially available instruments for comparison? (i.e., your Picarro G2401 )

**A comparison with different instrument has been added, although the technics and constraints of applications of these instruments can be very different (particularly for a Picarro analyzer).**

L200: Not essential, but I'm curious of you looked at the linear fit residuals. Was there an obvious curvature?

**No trend is being observed within the certified Picarro measurement range (below 20 ppm $CH_4$).**

L244: Missing work. Under the wind conditions…? (Check this sentence)

**I don't understand this comment.**

L270: What procedure do you use to interpolate/extrapolate your measurements to span x an z.

**More details have been provided to describe these steps. Integration along the y axis is performed with a simple linear interpolation, as the horizontal resolution of the measurement is very high. The interpolation of the measurements along the z axis between the horizontal transects. Measurements can also be extrapolated vertically if the lowest of highest horizontal transects do not reach the top or bottom of plume.**

L275: Section 3.2 would be a good place to include a description of the methodology to estimate flux errors. How do you assign uncertainties in your estimates based on the limitations of assumptions in this formulation (constant wind, homogeneous background, imperfectly known plume structure, etc)? What is the full error budget?

**More details have been given on the sources of uncertainties, although a complete uncertainty calculation is unfortunately still missing but will be the subject of future work.**

L325: Clarify what it is that the error is related to. Is this the estimated total emitted mass related to that recovered from measurements?

**The values presented were corresponding to the average of all relative errors of all quantifications, but this was not a good indicator and has been this has been removed.**

L325: Similarly, there is a need to quote uncertainty here. You have an estimate of "q" but the confidence levels for that estimate are not known. (I would expect that only a fraction on the total uncertainty is arising from the precision limitations of the instruments. Is this true?)

**More details on the influence of different sources of uncertainties (among which the instrumental uncertainties) are provided in the revised manuscript.**

L345: You mean modify, rather than modulate.

**Indeed. This has been corrected.**

L346: plan, not plane. Missing word: "allows *the sampling strategy* to adapt…"

**Ok. The sentence has been simplified.**

L 355: Ultrasonic (no "s")

**Ok, this has been corrected.**

L365: Mention of higher uncertainty. I agree this should be the case. However, a budget calculation performed here for your set up would be useful to show/prove this.

**Same answer as L275.**

L370: "Our flux estimates…" (not performances)

**Ok, this has been corrected.**

L370. You mean state of the art. However, you might say "commonly used methods"

**Indeed. This has been corrected.**

L403: This seems a major limitation. It would be useful to estimate the magnitude of the flux error associated wit this. What would you expect if you had vertical wind information? Can you illustrate this using the earlier data with the doppler lidar wind data?

**This would be a major limitation for field applications with sources at high elevation. But in the case our validation experiments during the TADI campaign, the use of a LiDAR wasn't the most appropriate wind monitoring method as well, since most of our measurement were performed at low elevation, below the LiDAR elevation range and thus the wind values had to be interpolated. This problematic is already discussed in the manuscript: multiple sonic anemometers distributed along a vertical mast would have been more appropriate in our case.**

L428: Here you certainly mean the simple mass balance method has a known set of causes of error. They can be large, due to the aspects of intermittency you imply.

**The reference emissions from the platform are indeed daily averaged values, which do not reflect the intermittency of the platform processes.**

L465: 7 flights were completed. (not exploited)

**Ok, this has been corrected.**

L482: These two sentences capture some discussion about possible sources of error, and is welcome. It needs to be expanded to give confidence in the characterization of error from the method as a whole, but particularly the impact of the new instrument measurement precision on the final uncertainty. I would guess it is not the leading source of error.

**More details have been given in terms of comparing the different sources of errors. The instrumental uncertainty is very small compared to the signal levels of our experiments. It is difficult to isolate the influence of all different sources of uncertainties based on our experiment: no direct link can be found between errors of the quantification and uncertainties associated with concentrations measurements or wind variability. It is possible that all these parameters play a combined role on the final quantification errors, and that they may be influence by other parameters more difficult to quantify. Therefore, we estimate that the results of these validation experiments in a whole are representative of the final uncertainty of our emissions quantification system.**

L515: It is claimed that the estimates are precise. This has not been shown. The sensor itself has a precision which is admirable, and needs to be compared to lab-based (etc) commercial instruments to show this. Even then, there is no assessment of how precise the flux estimates are following from the use of the UAV concentration measures within a simplified, and underconstrained mass balance calculation.

**This sentence has been modified. We agree that the method is lacking a complete estimation of uncertainties. This will be the subject of future improvements of the method.**

- **EC1**: 'Comment on amt-2022-334', Darin Toohey, 22 Jun 2023

Please address the comments of both reviewers in your revised paper, with particular emphasis on showing example(s) of observations and analysis of uncertainties in the flux estimates.

**A revised version of the manuscript will be provided, with more details on the analysis of different sources of uncertainties. An example of the observations during a flight of the TADI campaigns has also been included.**

**List of all changes made to the manuscript:**

*Line numbers are given relative the the track-changes file version of the new manuscript.*

- Title has been modified
- The abstract has been reworked

Introduction:

- Line 93: language corrections
- Lines 100-117: reworked details on the expected characteristics of mobile platforms for atmospheric concentrations monitoring
- Line 129: language correction
- Lines 138-145: reworked details of the expected precision and sensisty ranges of atmospheric concentration measurements
- Line 150: language correction
- Lines 160-164: gave more details about limitations of open cavity instruments
- Lines 166: removed $H_2O$ from the list of measured species by our instrument for more clarity as we don not perform spectroscopic $H_2O$ measurements
- Line 177: idem
- Line 190: language
- Lines 191-200: gave more technical details on the instrument
- Lines 202-207: reworked writing and introduction of the next sections
- Lines 215 and 2017: language
- Lines 226-231: Provided comparison with precisions of other types of analyzers
- Lines 295-296: provided more details of the Lidar instrument and introduced WD and WS
- Line 303: language
- Line 310-311: reworked to show that the method was already existing and included references to litterature
- Lines 318-319: added unit for concentration data and information that they must be expressed as dry basis volume concentration
- Lines 328-339: Reworked the description of the algorithm
- Lines 366-367: added missing information
- Lines 376-386: added description of new figures 6 and 7 showing raw values of one flight of a TADI controlled release experiment
- Lines 389-390: added details for each flight in the Supplementary Materials Tables S1 and S2
- Lines 400-425: modified description of results of the TADI experiments in terms of performance
- Lines 426-492: added section on the uncertainties of the method
- Lines 497-506: rephrased discussion on the detection limit of our method
- Lines 508-510: added information of the signal to noise ratio for the extrapolation of our results based on $CH_4$ to the quantification of $CO_2$
- Lines 511-521: new discussion on the uncertainties of the method
- Lines 522-540: rephrasing and more details on the technical limits of the LIDAR monitoring
- Lines 541-542: rephrasing
- Line 546: modified organization of the sub-sections
- Lines 548-549: rephrasing
- Lines 602-610: rephrasing
- Lines 615-617: added information on signal to noise ratio
- Lines 624-627:  added statistic on wind conditions during offshore platforms experiments
- Lines 652: rephrasing

- Lines 675-706: Modified what is presented as a main aspect of our study + rephrasing
- Lines 707-710: added information that uncertainties are determined empirically + rephrasing
- Lines 712-715: modified future perspectives
- Lines 723-724: rephrasing
- Figures 6 and 7: included new figures
- Tables 4 and 5: removed redundant information (unit in %)

---

## Author Response (AR2)

**Answers to the reviewers**

"Concerning the in-flight accuracy of the instrument, no experiment has been performed allowing an intercomparison with a reference instrument. This could be the subject of future work. " It will be useful to add the following sentence to the main text.

>> This information has been added in the conclusion of the manuscript, together with a new summary of the in-lab and in-flight precision of the instrument.

"These suggestions have been taken into account for the revision of the manuscript." It is rather difficult to judge to what degree these have been realized.

>> This refers to the following suggestions of RC2:

*"The paper presets results of source flux estimates but does not describe if these are useful. Are the estimates sufficiently robust (small enough error) to be of practical use? A discussion section (and possibly a motivating section in the introduction) that lays out the measurement need in terms of total error (perhaps a percentage or perhaps a total value) is needed.*

*During revision, the authors should look for clarity in what they wish to use as a title and what to emphasize in the abstract. At the outset, the paper has focus on instrumentation, but in the end claims the focus is on flux estimation technique, while the bulk of the results focus on case study applications. I caution that the lack of specific development of the estimation technique make the claim that there is method development difficult to argue. The bulk flux estimate from a mass balance method was not developed, and the error budget not described. While I do suggest the error budget be added during revision, it is also wise to ensure the text focuses toward the comparative natured of the results.*

*To this end care is needed as, for instance in the conclusion (L497-501) appears to suggest the technique itself was devised here. This is overstating the matter. Re-phrase sections like this to highlight that it is the combination of the sensor, the UAV platform, and the application of a mass balance method when used together that represent a measurement system. The essential new work done here, to my mind, is in all three areas: using a newly developed sensor designed for UAV work, using data from the sensor to drive a mass balance calculation (albeit, without error characterization), and then testing it in pragmatic cases that illustrate possible wide application."*

Concerning the first point, the referee asked for a discussion about the required precision of the estimates. We believe that the motivations for the development are already detailed in the introduction, showing that there is a need for top-down emissions monitoring technologies able to operate at the scale of an industrial site, at moderate costs and over short periods of time. A comparison with current top of the art technologies in the discussion section shows that our method is among the most precise methods to monitor this type of emissions scenarios. The achieved precision is currently sufficient for the needs of our partners in the oil and gas industry (TotalEnergies) who have chosen to apply it at a large scale to monitor the emissions of multiple sites worldwide.

Concerning the second point about what is emphasized as the main aspects of our study, it has been chosen to highlight the fact that the combination of the newly developed instrument, with a monitoring protocol and a data analysis approach constitutes a complete monitoring system which is the novelty of our approach. To this end, changes have been brought to the formulation of the title, the abstract has been reworked and changes have been brought to the conclusion. For instance, sentences describing the modelling approach as a development have been reworked, or the title now describes the method as a measurement system using UAV-based concentration monitoring.